# A coherent phonon-induced hidden quadrupolar ordered state in Ca$_2$RuO$_4$

Honglie Ning [1,2,6], Omar Mehio [1,2,6], Xinwei Li [1,2,6], Michael Buchhold [3], Mathias Driesse [2,5], Hengdi Zhao [4], Gang Cao [4] & David Hsieh [1,2] ✉

Ultrafast laser excitation provides a means to transiently realize long-range ordered electronic states of matter that are hidden in thermal equilibrium. Recently, this approach has unveiled a variety of thermally inaccessible ordered states in strongly correlated materials, including charge density wave, ferroelectric, magnetic, and intertwined charge-orbital ordered states. However, more exotic hidden states exhibiting higher multipolar ordering remain elusive owing to the challenge of directly manipulating and detecting them with light. Here we demonstrate a method to induce a dynamical transition from a thermally allowed to a thermally forbidden spin-orbit entangled quadrupolar ordered state in Ca$_2$RuO$_4$ by coherently exciting a phonon that is strongly coupled to the order parameter. Combining probe photon energy-resolved coherent phonon spectroscopy measurements with model Hamiltonian calculations, we show that the dynamical transition is manifested through anomalies in the temperature, pump excitation fluence, and probe photon energy dependence of the strongly coupled phonon. With this procedure, we introduce a general pathway to uncover hidden multipolar ordered states and to control their re-orientation on ultrashort timescales.

Photo-excited quantum materials can be transiently steered away from their ground state. Targeting nearby local minima in their potential energy landscapes may then reveal ordered states that are thermally inaccessible. This strategy has been adopted to drive dynamical transitions from structural[1–3], ferroelectric[4,5], magnetic[6,7], charge density wave ordered[8–10] and intertwined charge-orbital ordered[11] ground states into their metastable hidden counterparts. More exotic symmetry broken states that exhibit ordering of higher electronic multipole moments are prevalent in strongly interacting electron systems, notably the *f*- and heavy *d*-electron based materials[12–17]. However, because multipolar ordered states are challenging to directly manipulate and to detect using conventional techniques, experimental demonstrations of light-induced transitions between their equilibrium and hidden counterparts remain elusive.

Here we exploit the intrinsic coupling of a multipolar order parameter to lattice degree of freedom as a route both to impart and to detect a dynamical transition into a hidden state. We illustrate the general principle using a *d*-electron orbital polarized state as a representative example, which is described by an electric quadrupolar order parameter. Consider the general structural building block of a transition metal oxide, consisting of an oxygen octahedral cage surrounding a central transition metal ion with electrons occupying the $t_{2g}$ levels ($d_{xy}$, $d_{yz}$, $d_{xz}$). Above the critical temperature for quadrupolar order (QO), the system is characterized by a harmonic lattice Hamiltonian $\hat{H}_L = \sum_\gamma \frac{1}{2} B Q_\gamma^2$, where $B > 0$ is a constant and $\gamma$ runs over the orthornormal eigenmodes $Q_\gamma$ of the octahedral complex. The potential energy surface (PES) is a parabola with its minimum corresponding to a state with zero orbital polarization and zero octahedral distortion (Fig. 1a). Below the critical temperature, an additional Jahn-Teller (JT)

[1]Institute for Quantum Information and Matter, California Institute of Technology, Pasadena, CA, USA. [2]Department of Physics, California Institute of Technology, Pasadena, CA, USA. [3]Institut für Theoretische Physik, Universität zu Köln, Cologne, Germany. [4]Department of Physics, University of Colorado, Boulder, CO 80309, USA. [5]Present address: Institut für Physik und IRIS Adlershof, Humboldt-Universität zu Berlin, Zum Großen Windkanal, Berlin, Germany. [6]These authors contributed equally: Honglie Ning, Omar Mehio, Xinwei Li. ✉e-mail: dhsieh@caltech.edu

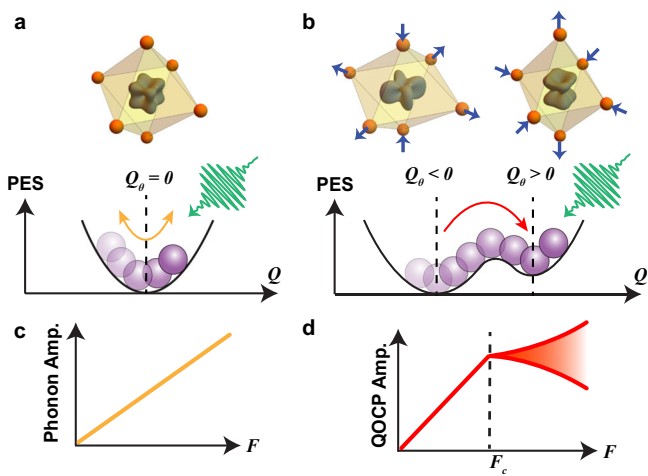

**Fig. 1 | Coherent phonon-based mechanism for accessing and detecting a hidden QO state. a** Schematic of PES above and (**b**), below $T_{QO}$. (Top) Electronic eigenstates and structural conformations of a transition metal oxide octahedral complex. Blue arrows point along the tetragonal distortion coordinate $Q_\theta$. **c** Pump fluence dependence of the coherent phonon amplitude expected for a conventional versus (**d**), QO-coupled phonon. Red shading depicts range of possible deviations from linearity.

term $\hat{H}_{JT} = \sum_\gamma g Q_\gamma \hat{\tau}_\gamma$ onsets, where $g$ is a coupling constant and $\hat{\tau}_\gamma$ is a quadrupolar operator that respects the same symmetry as $Q_\gamma$. The resultant PES possesses multiple local minima corresponding to QOs with different orbital polarizations and octahedral distortions. For the specific case of a tetragonal distortion $Q_\theta$ with $g > 0$ (Fig. 1b), the PES takes the form of a tilted double well with the ground (hidden) state corresponding to one with predominant $d_{xy}$ ($d_{xz/yz}$) orbital occupation and $Q_\theta < 0$ ($Q_\theta > 0$).

By optically exciting coherent atomic motion along the $Q_\theta$ phonon coordinate, the system can in principle overcome the potential barrier once a critical phonon amplitude is exceeded, allowing the system to transiently access the hidden state (Fig. 1b). At this transition, the excitation fluence ($F$) dependence of the quadrupolar order coupled phonon (QOCP) amplitude will exhibit an anomaly: in contrast to an ordinary uncoupled phonon whose amplitude continues to increase linearly with $F$ (Fig. 1c)[18], the amplitude of the QOCP will abruptly deviate from linearity due to the non-parabolicity of the PES (Fig. 1d). This unusual bifurcation in phonon dynamics serves as an experimental signature of dynamical switching between multipolar ordered ground and hidden states. This contrasts with how coherent phonons behave across light-induced order parameter melting transitions, which involves frequency softening and relaxation time divergence[1,19].

A well-suited experimental testbed is the $4d$-electron based multi-orbital Mott insulator $Ca_2RuO_4$, which is composed from a framework of corner-sharing oxygen octahedra, each surrounding a $t_{2g}^4$ ruthenium ion. Owing to a moderate intra-atomic spin-orbit coupling (SOC), the effective orbital ($L = 1$) and spin ($S = 1$) angular momenta of the two $t_{2g}$ holes combine into an ionic ground state with total angular momentum (pseudospin) $J = 0$[20,21]. However, inter-atomic superexchange interactions condense $J = 1$ spin-orbit excitons and drive the system into an antiferromagnetic state below a Néel temperature $T_N = 110$ K[22]. In a temperature window between $T_N$ and $T_{QO} = 260$ K, the system is reported to realize a spin-nematic state characterized by an electric quadrupolar order parameter, which is a spin-orbit coupled analog of an orbital ordered state[23] (Supplementary Note 1). However, this state has proven challenging to probe. Resonant x-ray diffraction experiments reveal only weak changes in the (100) and (013) Bragg peak intensities below $T_{QO}$, suggesting a small quadrupolar moment[22,24]. Moreover, no spatial symmetry breaking or structural anomalies have

been resolved below $T_{QO}$[24,25]. Nonetheless, a recent coherent phonon spectroscopy study showed that the 3.7 THz Raman active mode exhibits a $\pi$ phase shift across $T_{QO}$[26]. This suggests that the phonon is coupled to QO and may thus serve as the reporter for dynamical transitions into hidden QO states.

To search for signatures of a coherent phonon-induced hidden QO state, we performed pump fluence-dependent coherent phonon spectroscopy measurements on $Ca_2RuO_4$ single crystals. To coherently excite Raman-active phonons with large amplitude while minimizing heating due to absorption, the pump photon energy (0.3 eV) was tuned well below the low temperature Mott gap (~0.6 eV) and far away from phonon resonances (Supplementary Note 5)[27], allowing fluences up to 30 mJ/cm² to be reached. We found that the amplitudes of different coherent phonon modes vary strongly with probe photon energy. Therefore, we scanned the probe photon energy from 0.5 eV to 2.1 eV, covering the $d_{xy} \rightarrow d_{xz/yz}$ and the $d_{xz/yz} \rightarrow d_{xz/yz}$ absorption peaks near 1 eV and 2 eV, respectively[28,29], in order to capture a large set of modes.

## Results

Probe photon energy-resolved differential reflectivity transients measured at $T = 80$ K and $F = 15$ mJ/cm² are displayed in Fig. 2a. Coherent phonon oscillations are observed atop background exponential decay terms that describe the relaxation dynamics of charges excited via nonlinear absorption pathways, which has been studied elsewhere[30]. Upon subtracting the background term (Supplementary Note 2), beat patterns from multiple phonons become apparent (Fig. 2b). By applying a fast Fourier transform (FFT) to the background subtracted data, we were able to resolve a total of six Raman active phonon modes centered about 3.7, 5.7, 6.1, 7.5, 9.0, and 9.7 THz across the measured probe photon energy range (Fig. 2c), which are all assignable to $A_g$ modes based on previous Raman spectroscopy results[31]. A full set of such probe photon energy-dependent measurements was repeated for different pump fluences in order to track how the amplitude of each of the six modes, obtained through multi-Lorentzian fits to the FFT data, scales with fluence. Interestingly, the amplitude of the 3.7 THz mode abruptly deviates from linear scaling above a critical fluence $F_c$ ~15 mJ/cm², whereas the amplitude of all the other five modes continue to scale quasi-linearly up to $F = 25$ mJ/cm² at all the probe energies where they can be observed (Fig. 2d, e) (Supplementary Note 11). Absorption saturation at $F = 15$ mJ/cm² cannot explain this observation because that would cause a plateau in all mode amplitudes. A photo-thermal phase transition can be excluded because the 3.7 THz phonon drastically redshifts upon heating[26], whereas we observe negligible frequency shift across our entire measured fluence range (Supplementary Note 4). A non-thermal Mott insulator-to-metal transition can also be ruled out based on our transient optical conductivity data, which shows negligible spectral weight transfer into the Mott gap following the excitation (Supplementary Note 5). Instead, these behaviors are consistent with the 3.7 THz mode being an order parameter coupled phonon that drives a dynamical transition to a hidden state.

To determine whether the anomalous behavior of the 3.7 THz phonon exists only in the presence of QO, we tracked the fluence dependence of its amplitude as a function of temperature across $T_{QO}$ with a probe photon energy of 1.55 eV. Three features that indicate strong coupling between the 3.7 THz mode and the QO parameter are uncovered. First, the abrupt deviation from linear fluence scaling across $F_c$ is no longer observed above $T_{QO}$ (Fig. 3a). Second, the amplitude of the 3.7 THz mode measured at a fixed $F = 15$ mJ/cm² exhibits a prominent upturn below $T_{QO}$ (Fig. 3b). This phenomenon, as reproduced by the microscopic Hamiltonian (Supplementary Note 9), can be well understood because the amplitude of the QOCP measured at $F \geq F_c$ and $T \leq T_{QO}$ reflects the critical phonon amplitude for transitioning to a hidden state, which is also a proxy for the

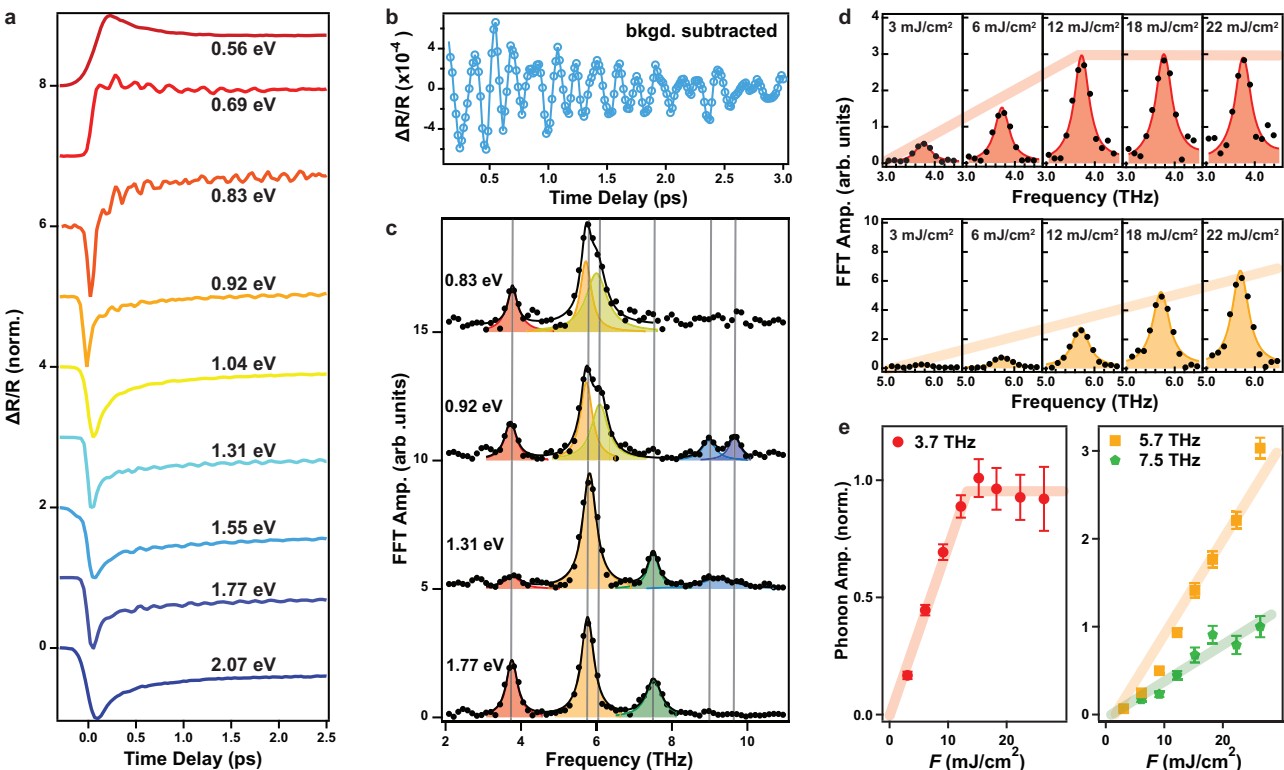

**Fig. 2 | Identification of the QOCP in Ca₂RuO₄. a** Normalized reflectivity transients $\Delta R/R$ pumped at 0.3 eV and $F = 15$ mJ/cm² and probed at select photon energies at $T = 80$ K. Curves are vertically offset for clarity. **b** Representative background-subtracted reflectivity transient probed at 1.55 eV. **c** FFT spectra of background-subtracted curves for select probe photon energies. Black lines show the best fits to a multi-Lorentzian function. Individual phonon modes are color-coded, with their center frequency marked by gray lines. Curves are vertically scaled and offset for clarity. The hump at 2.5 THz cannot be assigned to any known mode and may originate from PES anharmonicity (Supplementary Note 3). **d** Zoom-in on the 3.7 THz and 5.7 THz phonon peaks in the FFT spectrum probed at 1.55 eV for various pump fluences. Thin lines show Lorentzian fits. Thick lines are guides to the eye. **e** Pump fluence dependence of the amplitude of the phonons probed at 1.55 eV. All the data are normalized by the 3.7 THz phonon amplitude measured at $F = 15$ mJ/cm². Thick lines are guides to the eye. The error bars are obtained from the standard deviation of the multi-Lorentzian fitting to the FFT spectra.

maximal static JT distortion (inset of Fig. 3b). Increasing temperature reduces the magnitude of the distortion, bringing the minima of the PES closer to each other along the phonon coordinate $Q$ and thus decreasing the phonon amplitude. Third, $F_c$ decreases monotonically with increasing temperature and vanishes at $T_{QO}$ (Fig. 3c). This can be understood within our model because $F_c$ also marks the critical phonon amplitude for transitioning to a hidden state (Fig. 1). In this scenario, increasing temperature lowers the threshold coherent phonon amplitude for switching between different minima and thus decreases $F_c$. Above $T_{QO}$, the PES minima merge into a single parabola and therefore $F_c$ no longer exists. The absence of any discontinuity in $F_c$ or the phonon amplitude across $T_N$ further supports the picture of dynamical switching to a hidden QO state rather than to a hidden antiferromagnetic state.

We now try to develop a more quantitative theoretical understanding of the correlated dynamics of the QO and the coupled phonon. Based on the experimental observations, we can conclude that the QO transition is mainly coupled to the 3.7 THz phonon mode. This phonon can be represented as a superposition of several orthonormal octahedral eigenmodes $Q$. In transition metal oxides like Ca₂RuO₄, which possess only static $E_g$-symmetry distortions, the coupling between the $t_{2g}$ electrons and the tetragonal eigenmode $Q_\theta$, as well as the orthorhombic eigenmode $Q_\epsilon$, dominates the coupling to other eigenmodes (Supplementary Notes 12 and 13)[12,13,32,33]. To this end, we can construct a microscopic Hamiltonian for Ca₂RuO₄: $\hat{H} = \hat{H}_L + \hat{H}_{JT} + \hat{H}_U + \hat{H}_{SOC}$, which contains the previously introduced lattice and Jahn-Teller contributions $\hat{H}_L$ and $\hat{H}_{JT}$ but only including terms with $Q_\theta$ and $Q_\epsilon$ modes. To capture multi-orbital electronic interactions within the

$t_{2g}^4$ manifold, we include a generalized Kanamori-parametrized electronic correlation term $\hat{H}_U = (U - 3J_H)\frac{\hat{N}(\hat{N}-1)}{2} + \frac{5}{2}\hat{N} - J_H(2\hat{S}^2 + \hat{L}^2/2)$, where $J_H$ is the intra-atomic Hund's exchange, $U$ is the intra-orbital Coulomb interaction, $\hat{N}$ is the total number of electrons, and $\hat{L}$ and $\hat{S}$ are the total orbital and spin angular momentum operators, respectively (Methods and Supplementary Note 8)[34]. We also add a SOC term $\hat{H}_{SOC} = \lambda\hat{L} \cdot \hat{S}$, where $\lambda$ is the coupling strength, which endows the QO with spin-orbit entangled character. By performing exact diagonalization of the Hamiltonian $\hat{H}$ with microscopic parameters relevant for Ca₂RuO₄, we obtain the PES as a function of the two $E_g$ eigenmodes of the octahedron ($Q_\theta$ and $Q_\epsilon$). For $T < T_{QO}$, the PES exhibits three minima (Fig. 4a): the reported $d_{xy}$-dominant ground state with tetragonal compression along the $z$-axis[28,35], as well as two local minima with $d_{xz}$- or $d_{yz}$-dominant character and compression along the $y$- or $x$-axis, respectively. The latter two states are energetically degenerate and have slightly higher energy than the ground state, making them inaccessible in thermal equilibrium.

We then derive the equations of motion of ($Q_\theta$, $Q_\epsilon$) and solve for the system dynamics following an impulsive Gaussian pulse matching our experimental pump envelope (Methods and Supplementary Note 9). For small Gaussian peak heights, which correspond to $F < F_c$, the stimulus simply initiates oscillatory motion about the $d_{xy}$-dominant ground state minimum at the QOCP frequency. For $F > F_c$, the potential barrier is overcome and the hidden $d_{xz}$- or $d_{yz}$-dominant state is accessed. Depending on $F$ and the phonon damping rate (with fixed microscopic parameters), the system can then either directly relax to the local minimum, or continue to temporally oscillate back and forth between the global and local minima before eventually settling into

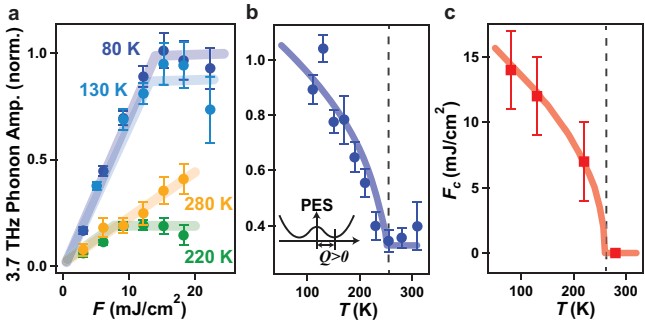

**Fig. 3 | Temperature dependence of the QOCP amplitude. a** Pump fluence dependence of the 3.7 THz phonon amplitude at select temperatures acquired with a 1.55 eV probe. Solid colored lines are guides to eye. All the data are normalized to the value measured at $F = 15$ mJ/cm$^2$ and $T = 80$ K. **b** Temperature dependence of the 3.7 THz phonon amplitude measured at $F = 15$ mJ/cm$^2$ with a 1.55 eV probe corrected for the pump-induced temperature rise (Supplementary Note 7) and (**c**) temperature dependence of the critical fluence $F_c$. The theoretical temperature dependence of the QOCP amplitude and $F_c$ calculated from our microscopic model are shown by the thick colored lines and are scaled vertically to match the experimental data (Supplementary Note 9). The dashed lines denote $T_{QO}$. The inset in (**b**) is a schematic showing that the phonon amplitude $Q$ measured at $F \geq F_c$ is a proxy for the maximal static JT distortion where the PES minimum is located. Note that the 3.7 THz phonon amplitude might exhibit an upturn above $T_{QO}$, which is not captured by our minimal model but has been observed elsewhere[26] (Supplementary Note 6). The error bars are obtained from the standard deviation of fitting to the data.

one of them. We calculated the temporal trajectory of $(Q_\theta, Q_\epsilon)$ for different fluences (Supplementary Note 9). The FFT of these projections were then used to determine the coherent oscillation amplitudes along the $Q_\theta$ and $Q_\epsilon$ coordinates (Fig. 4b). We find that the oscillation amplitudes along both coordinates scale linearly with fluence up to $F_c$ and then deviate, which is expected to be a general feature of any coherent phonon-induced dynamical transition (Fig. 1). A specific prediction of our model for Ca$_2$RuO$_4$ is that the deviation from linearity should be considerably more drastic for the $Q_\epsilon$ coordinate than for the $Q_\theta$ coordinate (Fig. 4b).

We propose the following experiment to test this prediction. Previous optical conductivity measurements report a peak ($\alpha$) between 0.5 and 1.3 eV that is attributed predominantly to the $d_{xy} \rightarrow d_{xz/yz}$ transition, and another peak ($\beta$) between 1.3 and 2.1 eV that is attributed predominantly to the $d_{xz/yz} \rightarrow d_{xz/yz}$ transition[27–29]. A probe photon energy tuned to the $\alpha$-peak is expected to be particularly sensitive to tetragonal distortions ($Q_\theta$) because this mode modulates the energy splitting between the $d_{xy}$ and $d_{xz/yz}$ levels (Fig. 4c insets). On the other hand, a probe photon energy tuned to the $\beta$-peak is expected to be particularly sensitive to orthorhombic distortions ($Q_\epsilon$), because this mode lifts the degeneracy of the $d_{xz/yz}$ bands and modulates their splitting. Since the eigenvector of the QOCP mode at 3.7 THz has a finite projection along both the $Q_\theta$ and $Q_\epsilon$ coordinates, our model predicts that the deviation from linear fluence scaling above $F_c$ observed for the 3.7 THz mode should be much more pronounced when the probe is tuned to the $\beta$-peak than when it is tuned to the $\alpha$-peak (Fig. 4b). Figure 4d shows the fluence dependence of the 3.7 THz phonon amplitude measured at $T = 80$ K using a series of probe photon energies. Remarkably, as the photon energy is tuned from the $\beta$ peak towards the $\alpha$ peak, the discontinuity at $F_c$ indeed becomes progressively less pronounced. This clearly distinct behavior of the $Q_\theta$ and $Q_\epsilon$ components of the QOCP serves as further evidence for a dynamical transition to a hidden QO state. The detailed nature of this hidden state, such as the ordering wave vector of the pseudospins, awaits further characterization using advanced ultrafast techniques such as time-resolved resonant x-ray scattering.

## Discussion

Altogether, these results demonstrate an out-of-equilibrium pathway to uncover thermally inaccessible QO states in Ca$_2$RuO$_4$ using coherent phonons. In tandem, we identify a set of highly unconventional coherent phonon properties that serve as unique signatures of transitions to the hidden states. Our protocol can be generally applied across strongly correlated materials to identify and manipulate exotic multipolar ordered states that typically elude conventional probes, with potential application to high-speed electronics beyond conventional spin- and charge-dipolar ordered materials. Future work using advanced time-dependent first-principles-based simulation tools can be employed to further elucidate the cooperative dynamics between the QO and lattice.

## Methods
### Sample preparation
High-quality plate-like single-crystal Ca$_2$RuO$_4$ with typical dimensions of 1 mm × 1 mm × 0.5 mm were obtained using a NEC optical floating zone furnace with control of growth environment[36]. The samples were cleaved along the (001) direction immediately before the experiment to obtain a fresh and smooth surface. The whole sample was then kept in high vacuum with pressure lower than $10^{-7}$ torr.

### Time-resolved broad-band reflectivity spectroscopy measurements
The sample temperature was fixed at 80 K without further notification. In the experimental setup, a Ti:sapphire amplified laser operating at 1 kHz generates 800 nm pulses with 40 fs time duration. Most of the power seeds two optical parametric amplifiers (OPAs) with individually tunable near infrared light ranging from 1200 to 2400 nm with a duration of 80 fs. One of the two OPAs pumps a differential frequency generator (DFG) and produces midinfrared light centered around 4000 nm with 100 fs time duration, which is used as the pump. The other OPA is used as a wavelength-tunable probe resonant with $d_{xy} \rightarrow d_{xz/yz}$ transition. To probe resonant with $d_{xz/yz} \rightarrow d_{xz/yz}$ transition, the remaining small portion of the 800 nm light and the second harmonic of the OPA light produced by BBO crystal are used. Our probe energy can thus scan from 0.5 eV to 2.1 eV. Si and InGaAs photodiodes are utilized as the photo-detectors. The fluence of the pump is tunable from 0 to 35 mJ/cm$^2$, above which thermal damage may occur as shown by a steady decrease of static reflectivity. The polarizations of the pump and the probe pulses were set to be perpendicular to each other, but both are parallel to the nearly isotropic [001] plane of the crystal.

### Many-body microscopic model
A comprehensive low energy microscopic model of Ca$_2$RuO$_4$ consists of four terms, and the detailed discussion on the cooperative SOC and JT effect can be found in Supplementary Note 8:

$$\hat{H} = \hat{H}_{SOC} + \hat{H}_{JT} + \hat{H}_L + \hat{H}_U, \tag{1}$$

and

$$
\begin{aligned}
\hat{H}_{SOC} &= \lambda \hat{\mathbf{L}} \cdot \hat{\mathbf{S}}, \\
\hat{H}_{JT} &= \sum_{\gamma = \theta, \epsilon} g Q_\gamma \hat{\tau}_\gamma, \\
\hat{H}_L &= \sum_{\gamma = \theta, \epsilon} \frac{1}{2} B Q_\gamma^2, \\
\hat{H}_U &= (U - 3J_H) \frac{\hat{N}(\hat{N} - 1)}{2} + \frac{5}{2} \hat{N} - J_H (2 \hat{S}^2 + \hat{L}^2 / 2).
\end{aligned}
\tag{2}
$$

The four terms correspond to SOC, JT interaction, lattice harmonic potential, and the multi-orbital electronic correlation. Here, $\lambda$ is the SOC constant, $g$ is the JT coupling constant, $B$ is the elastic lattice

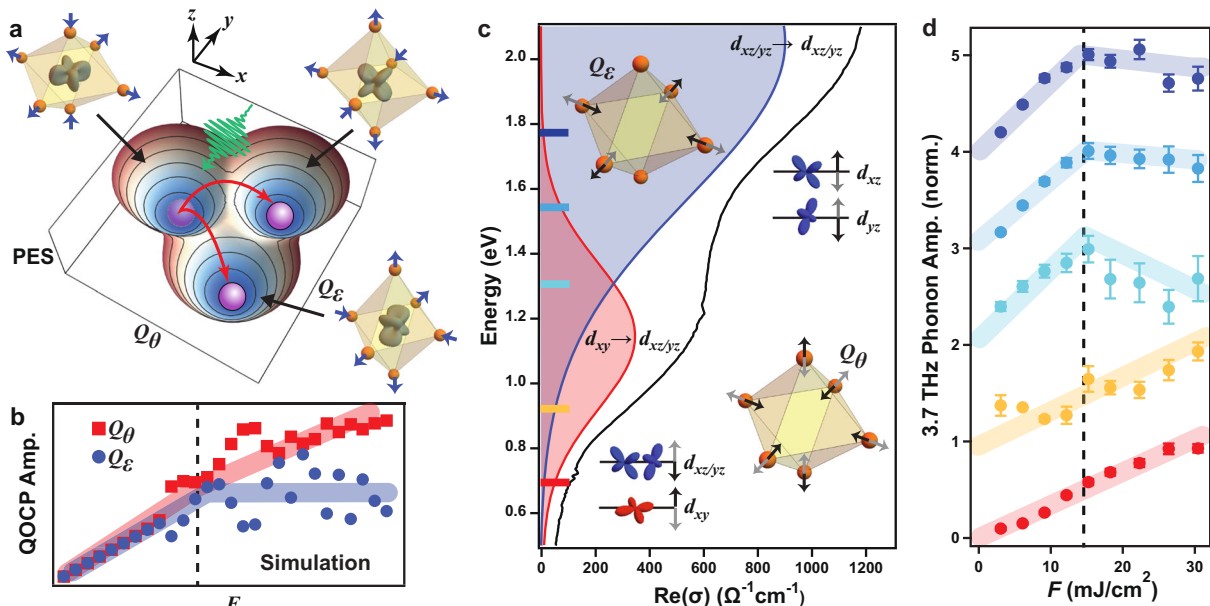

**Fig. 4 | Microscopic model of the hidden QO and signatures in orbital-selective coherent phonon spectroscopy. a** Calculated PES for $T < T_{QO}$ based on our microscopic model (Methods). Pseudospin distribution and octahedral distortion corresponding to each minimum is shown. Possible scenarios of collective lattice change are discussed in Supplementary Note 10. **b** Simulated pump fluence dependence of the QOCP amplitude projected along the $Q_\theta$ (red) and $Q_\varepsilon$ (blue) coordinates. Thick lines are guides to the eye and the dashed line denotes $F_c$. **c** Optical conductivity with Lorentzian fits corresponding to the $d_{xy} \rightarrow d_{xz/yz}$ (red) and $d_{xz/yz} \rightarrow d_{xz/yz}$ (blue) transitions[28]. Insets show the real space form of the $Q_\theta$ and $Q_\varepsilon$ distortions along with the induced modulation of orbital levels. **d** Pump fluence

dependence of the 3.7 THz phonon amplitude probed at 1.77, 1.55, 1.31, and 0.92, and 0.69 eV (see corresponding color bars in panel **c** and Supplementary Note 11). Data were acquired at $T = 80$ K. Thick lines are guides to the eye and the dashed line denotes $F_c$. Data are normalized to the maximal phonon amplitude measured at each corresponding probe energy and vertically offset for clarity. Note that the fluence dependence of the 3.7 THz mode with probe energies resonant with the $\alpha$-peak should not be confused with the linear fluence dependence of the uncoupled phonons. The nonlinear effects of $Q_\theta$ are relatively small and hardly resolved experimentally. The error bars are obtained from the standard deviation of the multi-Lorentzian fitting to the FFT spectra.

energy determined by the QOCP frequency, $J_H$ is the intra-atomic Hund's coupling, and $U$ is the Hubbard intra-orbital Coulomb interaction. $\hat{L}$ and $\hat{S}$ are the total orbital and spin angular momentum operators, $\hat{N}$ is the total number of electrons, and $\hat{\tau}$ is the quadrupolar operator which can be then written out as a linear superposition of the quadratures of angular momentum operators depending on the specific symmetry of the coupled eigenmodes $Q_\theta$ and $Q_\varepsilon$ (Supplementary Note 8). Since we are dealing with the isolated ion case with fixed number of electrons, we can safely set $U = 0$ and neglect the electron number operators, which only cause a constant energy shift[34]. We then performed exact diagonalization with electron number $N = 4$ and obtained the PES, which is the smallest eigenvalue of Eq. (1), as a function of the two structural order parameters $Q_\theta$ and $Q_\varepsilon$. A tetragonal splitting term $\hat{H}_{TS} = \Delta \hat{L}_z^2$, which is ignored in our minimal model, can be included to characterize the out-of-plane anisotropy, where $\Delta$ is the tetragonal splitting energy. This will elevate the $d_{yz}$ and $d_{xz}$ minima above the $d_{xy}$ minimum, so that $d_{xy}$-dominated QO is uniquely reached, which is indeed the case for Ca$_2$RuO$_4$.

## Dynamical simulation

We denote the smallest eigenvalue of Eq. (1), the ground state PES, as $V(Q_\theta, Q_\varepsilon)$. Based on the experimental observations (Supplementary Note 14), we assume an impulsive instead of displacive excitation occurs at time zero which initiates the time evolution of $Q_\theta$ and $Q_\varepsilon$. Since the pump light frequency is one order of magnitude larger than the frequency of any observed phonons, we average out the sinusoidal oscillatory part of the pump but retain its Gaussian envelop with a time duration $\sigma$ of 0.1 ps. The Gaussian amplitude is proportional to the pump fluence $F$ with a scaling factor $A$. We also include a phonon damping term with decay constant $\gamma$ which can be experimentally determined. Then the equations of motion of QOCP along $Q_\theta$ and $Q_\varepsilon$

coordinates can be expressed as:

$$\frac{d^2 Q_{\theta/\varepsilon}}{dt^2} + 2\gamma \frac{dQ_{\theta/\varepsilon}}{dt} + \frac{dV(Q_\theta, Q_\varepsilon)}{dQ_{\theta/\varepsilon}} = AF \exp\left[ -\frac{4\ln(2) t^2}{\sigma^2} \right]. \quad (3)$$

We determine all the parameters from X-ray scattering, optical spectroscopy, and angle-resolved photoemission spectroscopy results. Detailed simulation results and evidence for the insensitivity of our main conclusions to small changes of parameter values can be found in Supplementary Note 9.

## Density functional theory simulation

The optical conductivity spectra in the presence of specific lattice distortions were calculated from first principles using density functional theory (DFT) as implemented in the Quantum ESPRESSO package[37–39]. We used Perdew-Burke-Ernzerhof (PBE) functionals along with fully-relativistic Optimized Norm-Conserving Vanderbilt Pseudo-potentials (ONCVPSP). The crystal structure of Ca$_2$RuO$_4$ was obtained using the 90 K data from ref. 24. For self-consistency, the energy cutoff and grid density are converged with respect to total energy, with wavefunction cutoff of 50 Ry and a $6 \times 6 \times 3$ Monkhorst-Pack grid as final values. For electronic structure calculations, an additional non-self-consistent calculation was performed at higher k-mesh of $9 \times 9 \times 4$. Following ref. 30, we included collinear spin polarization and DFT + $U$ with $U = 3.5$ eV. Simulations of the optical spectrum used the Epsilon.x code to smooth out numerical instabilities (interband smearing). We used the Phonopy package to calculate the phonon eigenmodes and frequencies under the frozen-phonon approximation using a $2 \times 2 \times 1$ supercell, allowing for dimensions > 10 Å along each axis[40]. We analyzed the effect of the 3.7 and 5.7 THz phonon modes with atomic

displacements on the order of 0.01 Å for oxygen atoms and 0.1 Å for calcium atoms to avoid nonlinear effects.

## Data availability

Source data are provided with this paper. All other data that support the findings of this study are available from the corresponding author upon request. Source data are provided with this paper.

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

## Acknowledgements

We thank Min-Cheol Lee and Jong-Seok Lee for sharing optical conductivity data. Time-resolved optical spectroscopy measurements and model Hamiltonian calculations were supported by an ARO PECASE Award No. W911NF-17-1-0204. D.H. acknowledges support for instrumentation from the David and Lucile Packard Foundation and the Institute for Quantum Information and Matter, an NSF Physics Frontiers Center (PHY-1733907). M.B. acknowledges support from the Deutsche Forschungsgemeinschaft (DFG, German Research Foundation) under Germany's Excellence Strategy Cluster of Excellence Matter and Light for Quantum Computing (ML4Q) EXC 2004/1 390534769, and by the DFG Collaborative Research Center (CRC) 1238 (Project No. 277146847, subproject C04) and CRC 183 (Project No. 277101999, subproject B02). Work at C.U. Boulder is supported by the National Science Foundation via Grant No. DMR 2204811.

## Author contributions

H.N., X.L., O.M., and D.H. conceived the experiment. X.L. and H.N. performed the coherent phonon spectroscopy measurements. H.N. and M.B. developed the many-body theory and conducted the dynamical simulations. H.N., X.L, and O.M. analyzed the data. M.D. conducted the DFT simulations. H.Z. and G.C. synthesized and characterized the samples. H.N. and D.H. wrote the manuscript with input from all authors.

## Competing interests

The authors declare no competing interests.
