## [Peer Review File · Nature Communications]

REVIEWER COMMENTS

Reviewer #1 (Remarks to the Author):

The authors report ultrafast responses of Ca_2RuO_4 , suggesting that a hidden quadruple order can be studied by coherent oscillations coupled to the order. I find their experimental results interesting revealing a coupling of the phonon mode with a transient order. But there are quite some points that are not clear yet.

1. The authors mentioned that the pumping impulsively excites the system. Impulsive excitation results in a sin-type coherent oscillations and usually observed in a transparent material. Although the pump photon energy is lower than the lowest excitation peak and extrapolated energy gap of the peak, more evidence has to be provided to claim an impulsive excitation.

2. They suggested that at high fluence above F_c the system may transiently enter a hidden quadruple state stabilizing dx_z and/or dy_z state. Through the manuscript, it is not clear what is driving this phase transition. If it were the 3.7 THz phonon mode that is impulsively excited by pumping, then a detailed explanation is desired why the lattice mode can drive the order. If the phonon eigen mode could be composed with two lattice deformations Q_θ and Q_ϵ they discussed, then the phonon might provide some potential change for the hidden order. However, as far as I see, the phonon eigen mode is not a superposition of the two lattice deformations and therefore, I don't find a ground for their explanation.

3. Based on the linear fluence dependence of the 3.7 THz mode observed in the low probe photon energy region, I expect that the phonon amplitude increases linearly with the fluence even up to the highest fluence in their measurement. If it were impulsive excitation, the oscillation will show a sin-like phase at least in the low fluence limit. On the other hand, if the potential energy surface of the lattice deformation Q of the phonon is directly coupled to the hidden order as shown in Fig. 1b, then the oscillation amplitude should deviate from the linear response as depicted in Fig. 1d. That is, the linear increase of the phonon amplitude in Fig. 4d and the model in Fig. 1 cannot compromise with each other.

Instead, if the 3.7 THz mode were not directly coupled to the quadruple (or any other) order but the optical modulations due to the phonon oscillations should depend on the order parameter, then their observations could be consistent with each other. However, the origin/driving force of the new order still remains unclear.

4. If the phonon mode is strongly coupled to the order parameter as discussed in Fig. 1, then the oscillation phase is also expected to deviate from the sine-like phase near F_c . Indeed, the reference #26 reported that the oscillation phase changes by 180 degrees depending on temperature. The temperature is not mentioned but figure S2 implies that some oscillation phase may change

depending on fluence and/or photon energy. To understand better, it is desired to discuss the fluence dependent evolution of the oscillation phases.

5. Figure 3b shows temperature dependence of the phonon amplitude. Their data at room temperature somehow have a very large error bar, resulting in practically no change above $T_{QO} \sim 260$ K. However, the reference #26 shows that the oscillation amplitude increases above 260 K. It is desired to explain the difference between these two data. It seems that the error bar becomes relatively larger at higher fluence according to Fig. 3. So, temperature dependence measured at lower pump fluence should be helpful to understand the evolution better. In addition, temperature dependent oscillation amplitude at different probe energies (near peak alpha and beta) is also desired to clarify the least coupling of the phonon mode with the alpha peak.

6. In the simulation, they initiated Q_{θ} and/or Q_{ϵ} instead of the phonon mode obtained from their DFT calculations and obtained the oscillation amplitude. What are the oscillation frequencies observed in the simulation? Do they result in the same oscillation frequency? I suppose that Q_{θ} and Q_{ϵ} displacement cannot oscillate at the observed phonon frequency. Simulations for the observed phonon mode depending on fluence are desired instead of the Q_{θ} and Q_{ϵ} .

I believe that their observations could indeed be related with some transient ordering. With above mentioned points, however, I think it is not clarified well enough why the ordering has to be the quadruple order. Therefore, I cannot recommend the publication until they clarify all the points.

As a minor point, it is desired to mention important experimental conditions in more details, such as temperature and photon energy, in Fig. 4, Fig. S2.

Reviewer #2 (Remarks to the Author):

The authors have presented a very comprehensive and solid study of the coherent phonon induced ordered state in Ca_2RuO_4 . I find the results very interesting and important, since the observation of the hidden order in Ca_2RuO_4 , as the authors written in the text, was quite challenging to be realized. Thus, I recommend to accept this paper for publication in Nature communications after the following suggestions are considered:

1. The hidden quadrupolar order observed by the authors was attributed to an orbital ordered state occurs at the tetragonal phase transition point $T_{QO} = 260\text{K}$. If I understand correctly, below $T_N = 110\text{K}$

where the long range magnetic order forms, the system should correspond to a dipolar order which has orthorhombic symmetry. Then why do the observed features at $T=80$ K (below T_N) almost identical to those at $T=220$ K (between T_N and T_{QO})?

2. Section 1 in the supplementary materials is a little bit confusing and needs to be improved. For instance:

a) what do the two shapes on t_{2g} orbitals in Fig. S1 represent?

b) Notations of “ mJ ” with capital “ J ” in the figure while it is small “ j ” in the text.

c) “Due to quasi-2D XY-type fluctuations, the magnetic ordering temperature T_N is lowered [11]”, I don't think “ T_N is lowered” “Due to quasi-2D XY-type fluctuations” is concluded by Ref.11. What Ref.11 delivers was the magnetic order should coincide with the lattice symmetry: quadrupolar order for tetragonal lattice and dipolar order for orthorhombic lattice.

Reviewer #3 (Remarks to the Author):

Review: A coherent phonon-induced hidden quadrupolar ordered state in Ca_2RuO_4

The authors describe an experiment in which below-gap, mid-infrared light is used to impulsively excite various Raman-active modes in the Mott insulator Ca_2RuO_4 . The anomalous dependence of one phonon amplitude on pump fluence, temperature, and probe photon energy suggests that the material has transitioned dynamically to a thermally-inaccessible quadrupolar order. The stark disparity in the behavior of the quadrupolar order coupled phonon (QOCP) compared to the other observed coherent modes supports the authors' interpretation. While the paper does not introduce a novel approach to detecting quadrupolar orders (QOs), it does demonstrate how a transition among different QOs can be induced and experimentally measured.

In principle, I think that the idea is novel and that the model is solid. However, I have both major and minor concerns regarding the paper that the authors should address before I can recommend it for publication.

1. One of the most critical observations is that the QOCP amplitude saturates with fluence, while the amplitude of the other modes scales linearly with it. Regrettably, the figure (Fig. 2e) intended to confirm this behavior is confusing to me. Specifically, I believe that overlaying the data for all other

phonons (Fig2e, right) could be misleading. Although I think a linear fit is appropriate for the 5.7 THz phonon, the fluence dependence of the 6.1 and 7.5 THz phonons appears to exhibit a similar, although less evident, saturation behavior at approximately 17.5 mJ/cm^2 .

To unequivocally demonstrate that the anomalies concern only the 3.7 THz phonon, I propose the following suggestions:

1.1 - Fig.2e should display the fluence dependence for all phonons separately, similar to what is done for the 3.7 THz phonon. If possible, more data points (finer fluence steps) should be measured to differentiate between noise on top of a linear dependence and an actual saturation behavior for the other phonons.

1.2 - The contents of Fig.3 should be presented in the supplementary materials for all the other phonons as well. Specifically, how does the amplitude of the other phonons change as a function of temperature?

1.3 – Also the contents of Fig.4d should be shown for all the other phonons. Does the fluence dependence remain linear at all other probe photon energies?

Most of the data I am requesting the authors to present should be readily available from the data sets they have already measured.

2. The paper lacks clarity on how the driven Ag phonons affect and relate to the Eg tetragonal and orthorhombic deformations Q_{θ} and Q_{ϵ} are excited. The authors should explain whether the eigenvector of the 3.7 THz mode is mapped onto the Q_{θ} and Q_{ϵ} coordinates and how. Additionally, it is important to know whether the displacements of the other Ag modes at higher frequencies contribute to the Q_{θ} and Q_{ϵ} deformations and if this was considered in the calculations. In general, the authors should provide a better explanation of the relationship between the driven coherent phonons and the deformations used in their calculations.

3. The critical displacement (in pm) needed to transition to the hidden state should be estimated from both the experiments and calculations. This information can be valuable and could also be measured with the x-ray scattering experiments that the authors already considered.

4. The paper states that the pump photon energy (0.3 eV) was chosen to be less than the gap (0.6 eV) and far away from phonon resonances. However, the latter information should be mentioned in the main text and not just in the supplementary material. It would also be helpful if the authors

could provide a reference for or show the optical properties of the material at the pump wavelength, such as its absorption, reflectivity, transmission, etc.

5. The same group has published and cited a previous work where they study the doublon-holon pair production in the same material and under the same excitation conditions [30]. Have the authors considered how such a simultaneous nonlinear process influences/affects the transition to the hidden state?

6. In the supplementary material, Fig.S6e, Fig.S7a,b,c,d,e,f, Fig.S9c are missing.

7. The authors should make the phonon amplitudes consistent for comparison. For instance, from Fig.3b, it is expected that the phonon amplitude at 15 mJ/cm^2 would increase by approximately 2.5 times going from $\sim 280\text{K}$ to $\sim 80\text{K}$. However, in Fig.3a, the phonon amplitude at these two temperatures and at 15 mJ/cm^2 is the same. It is unclear whether the data in Fig.3a is normalized. The authors should normalize the data when necessary and show the fit values otherwise to enable comparison between different data sets. The same applies to Fig.2e, where it is unclear whether the data on the right is normalized and the data on the left is not.

8. Fig.3a would benefit significantly from showing the data in the same plot without normalization. This would enable visualization of the changing critical fluence, maximum amplitude, and shape, as well as the stark increase in photo-susceptibility (the slope of the fluence dependence before saturation) for decreasing temperatures.

Response to Reviewer #1

The authors report ultrafast responses of Ca₂RuO₄, suggesting that a hidden quadruple order can be studied by coherent oscillations coupled to the order. I find their experimental results interesting revealing a coupling of the phonon mode with a transient order. But there are quite some points that are not clear yet.

We thank the Reviewer for finding our work interesting. We also thank the Reviewer for providing valuable comments. Below please find our response to each comment. **The resulting actions taken are summarized and noted in bolded text.**

1. The authors mentioned that the pumping impulsively excites the system. Impulsive excitation results in a sin-type coherent oscillations and usually observed in a transparent material. Although the pump photon energy is lower than the lowest excitation peak and extrapolated energy gap of the peak, more evidence has to be provided to claim an impulsive excitation.

As mentioned by the Reviewer, impulsive stimulated Raman scattering (ISRS) occurs in the transparent regime, exciting coherent phonons with a sinusoidal oscillation. On the other hand, displacive excitation of coherent phonons (DECP) relies on real electronic excitations, which modifies the potential energy surface (PES) and dominates the absorbing regime, resulting in a cosinusoidal oscillation of the phonons. To ensure predominantly impulsive excitation, we choose a pump photon energy well below the Mott gap of Ca₂RuO₄ and far away from phonon resonances to maximize (minimize) transmission (absorption coefficient). Our selection is supported by previous optical spectroscopy measurements [*Solid State Communications* **133**, 103–107 (2005); *Phys. Rev. Lett.* **91**, 056403 (2003)], demonstrating at least one order of magnitude higher transmission and nearly 16 times smaller absorption coefficient at our pump energy (0.3 eV), as compared to 1.1 eV where the first absorption peak occurs.

The assumption that the quadrupolar-order-coupled phonon (QOCP) is impulsively excited is further justified by two observations. First, the phase of the 3.7 THz mode is close to $\pm\pi/2$. We directly complex Fourier transform the background-subtracted transient reflectivity traces at different probe energies. Assuming a damped cosinusoidal function $A\exp(-t/\tau)\cos(\omega t+\varphi)$, where A , τ , ω , and φ are the amplitude, lifetime, frequency, and phase of the phonon, the phase φ can be directly obtained by simultaneously comparing the real and imaginary part of the FFT spectra as shown by the schematics in Fig. R1a. Our results show that the FFT spectra of the QOCP better

match the spectra of $\varphi = (\pm)\pi/2$ at probe energies resonant with both the α -peak (0.92 eV) and β -peak (1.77 eV), corroborating the ISRS mechanism (Fig. R1b).

Fig. R1. **a**, Schematics of real and imaginary FFT spectra of damped cosinusoidal functions of different phases as denoted. **b**, Complex FFT spectra measured with a pump fluence of 9 mJ/cm^2 and 1.77 eV and 0.92 eV probes. Thin lines indicate the phonon positions. **c**, Fluence dependence of the amplitude of $\Delta R/R(t=0)$, the 3.7 THz phonon, and the 5.7 THz phonon measured at 1.77 eV probe. Gray lines are linear and quadratic fits.

Second, the linear pump fluence dependence of the QOCP amplitude before saturation argues against the prediction of the DECP mechanism but matches the prediction of the ISRS mechanism. The DECP mechanism predicts a phonon amplitude proportional to the carrier density n . In the above-gap pumping case, carriers are generated through linear absorption and n is proportional to F ; in the subgap pumping case, carriers are nonlinearly generated through either multi-photon absorption or quantum tunneling processes, which gives rise to a superlinear F -dependence of n . Previous work on Ca_2RuO_4 already showed that subgap pumping leads to nonlinear carrier generation [*Phys. Rev. Lett.* **128**, 187402 (2022)], manifested through a superlinear fluence dependence of $\Delta R/R(t=0)$, which is proportional to n (Fig. R1c). Therefore, if the QOCP is launched displacively, its amplitude should scale superlinearly with F like $\Delta R/R(t=0)$. However, we observed a linear fluence dependence of the QOCP before its amplitude saturates, thus refuting the DECP scenario. On the other hand, an impulsively excited phonon is stimulated through a two-photon Raman scattering process, whose amplitude is proportional to the square of the electric field strength E^2 (i.e. F), independent of the pump photon energy. This matches our observation. Consequently, the evidence once again supports the dominance of the ISRS mechanism over DECP. We note that the above arguments can also apply to the QO-uncoupled modes (Fig. R1). **We have now incorporated the aforementioned discussions into a new SI Section 14 elaborating on the dominant impulsive excitation mechanism of the QOCP.**

Nevertheless, it is important to acknowledge that despite the QOCP being predominantly impulsively excited, we cannot completely exclude a small finite contribution of DECP. To account for this, **we have revised our language about the driving mechanism in the main text, stating that we "coherently" instead of "impulsively" drive the phonons.** It is crucial to note

that even if the QOCP is displacively excited, the switch of hidden quadrupolar order (QO) would still occur and our main conclusions would remain unchanged. This is because the QO switch is induced by the coherent phonon oscillation and is independent of the specific mechanism that drives the phonon at $t = 0$. **We have also included this point in SI Section 14 to emphasize the robustness of the QO switch.**

2. They suggested that at high fluence above F_c the system may transiently enter a hidden quadruple state stabilizing dxz and/or dyz state. Through the manuscript, it is not clear what is driving this phase transition. If it were the 3.7 THz phonon mode that is impulsively excited by pumping, then a detailed explanation is desired why the lattice mode can drive the order. If the phonon eigen mode could be composed with two lattice deformations Q_θ and Q_ϵ they discussed, then the phonon might provide some potential change for the hidden order. However, as far as I see, the phonon eigen mode is not a superposition of the two lattice deformations and therefore, I don't find a ground for their explanation.

As suggested by the Reviewer, the QO transition is indeed driven by the 3.7 THz phonon mode, which can be mapped onto the two octahedral eigen-deformations Q_θ and Q_ϵ . This mapping theoretically applies to all phonon eigenmodes. Considering a perfect octahedron with O_h point group symmetry, any inversion-symmetric distortion can be represented as a superposition of eigen-distortions respecting A_{1g} , A_{2g} , E_g , T_{1g} or T_{2g} symmetries. Therefore, these eigenmodes form a complete orthonormal basis for any type of distortion of an isolated octahedron. For the t_{2g}

Fig. R2. Schematics of the eigenmodes of octahedron that can couple to t_{2g} electrons. **a**, Q_A , **b**, Q_{E0} , **c**, Q_{Ee} , **d**, $Q_{T\zeta}$, **e**, $Q_{T\eta}$, and **f**, $Q_{T\zeta}$. The red balls represent O atoms at the apices of the octahedra and the gray balls represent Ru atoms.

manifold electrons, only six different distortions respecting A_{1g} , E_g ($Q_{E\theta}$, $Q_{E\varepsilon}$), and T_{2g} ($Q_{T\eta}$, $Q_{T\xi}$, $Q_{T\zeta}$) symmetries are allowed to couple to the electrons, whose spatial configurations are displayed in Fig. S9 as reproduced here in Fig. R2 [*The Jahn-Teller Effect* (Cambridge University Press, Cambridge, 2006)].

In other words, any t_{2g} -electron-coupled phonon eigenmodes, regardless of their symmetries with respect to the entire lattice, is a superposition of the six octahedral eigenfunctions: $Q = a_A Q_A + a_\theta Q_{E\theta} + a_\varepsilon Q_{E\varepsilon} + a_\xi Q_{T\xi} + a_\eta Q_{T\eta} + a_\zeta Q_{T\zeta}$, where the a 's represent the ratio of different components. Since Q_A is fully symmetric and thus Jahn-Teller (JT) inactive, the leading-order bilinear JT coupling term can be expressed as: $g_E(a_\theta Q_{E\theta} \tau_{E\theta} + a_\varepsilon Q_{E\varepsilon} \tau_{E\varepsilon}) + g_T(a_\xi Q_{T\xi} \tau_{T\xi} + a_\eta Q_{T\eta} \tau_{T\eta} + a_\zeta Q_{T\zeta} \tau_{T\zeta})$, where g 's are the JT coupling constants and τ 's are the electronic QO parameters. In transition metal oxides with only static E_g (tetragonal, orthorhombic) distortions like Ca_2RuO_4 , the T_{2g} terms can be omitted because g_T is significantly smaller than g_E [*Phys. Rev. B* **98**, 075138 (2018), *Phys. Rev. Lett.* **116**, 106401 (2016)]. Consequently, the coupling strength between the QO and any phonon mode is determined by $g(a_\theta Q_{E\theta} + a_\varepsilon Q_{E\varepsilon})$ (subscript “E” is dropped for simplicity). While these concepts have been introduced in Supplementary Section 7, **we now added a new paragraph in the main text to highlight the relationship between the driven coherent phonons and the octahedral deformations used in their calculations. Moreover, we reorganized a more detailed version of these important discussions in a new Supplementary Section 13.**

Having established the theoretical foundation, immediate questions are how the 3.7 THz QOCP is mapped to Q_θ and Q_ε and why the other modes are not coupled to QO. To address these questions, **we utilized two methods to demonstrate how the 3.7 THz mode can be mapped onto Q_θ and Q_ε , i.e. obtain a_θ and a_ε . Furthermore, we applied the same analysis to the 5.7 THz phonon as a comparison, which is an unambiguously QO-uncoupled mode due to its consistent linear fluence dependence across the entire probe energy range.**

First, we qualitatively analyzed the components of Q_θ and Q_ε , i.e. a_θ and a_ε , by examining the amplitude and direction of the phonon eigenvectors. Ca_2RuO_4 is composed of in-plane corner-shared octahedra. As the octahedra undergo distortion, a collective tilting or rotation must follow to accommodate the bond length changes. By considering these geometric constraints, we imposed the required length change in the Ru-O bond of Q_θ and Q_ε and simulated the eigenvectors of Q_θ and Q_ε within the entire lattice after accommodating the bond length changes. Interestingly, we find that the eigenvector of the tetragonal distortion Q_θ mainly exhibits an out-of-plane component, indicating a collective tilting of the octahedra along an in-plane axis (Figs. R3a-c). On the other hand, the eigenvector of the orthorhombic distortion Q_ε mostly shows an in-plane component, suggesting a collective rotation of all the octahedra along an out-of-plane axis through the center Ru atoms (Figs. R3d-f). As a result, analyzing the direction of the eigenvectors allows us to roughly distinguish the components of Q_θ and Q_ε for different phonons. Another distinguishing feature between the two modes is the amplitude of displacement of the apical oxygens. The length of the apical Ru-O bonds remains unchanged in the Q_ε mode, so the apical oxygen atoms only displace slightly following the rotation of the octahedron. However, the net displacement of the apical

oxygen of the Q_θ mode is significantly larger, due to the combination of tilting of the octahedron and the required length change for Q_θ .

Fig. R3. **a**, Schematic of the tetragonal distortion in the lattice. Only one layer of Ru-O octahedra is shown for clarity. Black arrows mark the desired displacement of the planar oxygen atoms, which are shared by two octahedra, in each individual octahedron. **b**, Schematic of the eigenvector of the tetragonal distortion in the lattice as marked by the yellow arrows, which are the sum of black arrows at each atomic position. **c**, Schematics showing the out-of-plane tilting of octahedra induced by the tetragonal distortion from the top and side views. Dashed lines indicate the tilting axes. **d-f**, Same as **a-c** but for the orthorhombic distortion.

We now examine the eigenvectors of the 3.7 THz and 5.7 THz modes obtained from our DFT simulations (Supplementary Section 12) using the aforementioned two criteria. We find that the eigenvectors of the 3.7 THz mode displays a notably larger in-plane displacement of the planar oxygens and a larger displacement of the apical oxygens compared to the 5.7 THz mode. These two observations suggest a larger component of both Q_ϵ and Q_θ components in the 3.7 THz mode compared to the QO-uncoupled 5.7 THz mode. Given the larger projection on Q_ϵ and Q_θ , the 3.7 THz mode appears to be a more suitable candidate for QOCP than the 5.7 THz mode.

To more quantitatively obtain a_θ and a_ϵ , we calculated the length change in the four planar Ru-O bonds and two apical Ru-O bonds based on the DFT results. Theoretically, Q_θ induces a length change with the same amplitude (d_θ) and sign in the four planar Ru-O bonds, while Q_ϵ results in a length change with the same amplitude (d_ϵ) and opposite signs in the neighboring planar Ru-O

bonds. Consequently, the net length change of two neighboring planar Ru-O bonds should be d_θ - d_ϵ and $d_\theta+d_\epsilon$ when both distortions are present. The apical Ru-O bond change, on the other hand, predominantly originates from Q_θ . Since $d_\theta:d_\epsilon=a_\theta:a_\epsilon$, analyzing the length change of different Ru-O bonds provides a direct quantitative way to determine a_ϵ and a_θ . With identical Ca atom displacements for both modes, we find that the neighboring planar Ru-O bond length change ratio is roughly 1:2.36 and 1.29:1.32 for the 3.7 THz and 5.7 THz modes, respectively. This indicates that $a_\theta:a_\epsilon=1.68:0.68$ and $1.31:0.01$ for the two modes, respectively. In other words, the projection on $Q_\theta(a_\theta)$ of the 3.7 THz mode is nearly $1.68/1.31=1.3$ times higher than that of the 5.7 THz mode. This is further corroborated by the length change in the apical Ru-O bond, with the 3.7 THz mode exhibiting nearly double the bond length change compared to the 5.7 THz mode. On the other hand, the projection on $Q_\epsilon(a_\epsilon)$ of the 5.7 THz mode is nearly zero. The negligible composition of Q_ϵ in the 5.7 THz mode explains the absence of phonon amplitude saturation (which is most apparent in Q_ϵ) in the 5.7 THz mode.

Based on the aforementioned qualitative and quantitative analysis, we have demonstrated that both a_θ and a_ϵ are higher in the 3.7 THz mode. However, the formula $g(a_\theta Q_\theta \tau_\theta + a_\epsilon Q_\epsilon \tau_\epsilon)$ indicates that the coupling strength to the QO is also determined by g in addition to a_θ and a_ϵ . Interestingly, our DFT simulation (Fig. S18c) has shown that the 3.7 THz mode induces a larger change in optical conductivity than the 5.5 THz mode at the probe energies sensitive to both Q_θ and Q_ϵ modes. This result indicates a larger g of the 3.7 THz mode, further supporting that the 3.7 THz mode is strongly coupled to the QO. As a result, our comprehensive analysis not only reveals how the phonons are projected onto Q_ϵ and Q_θ but also explains why the 3.7 THz mode is more strongly coupled to the QO. **We have now included all these discussions in the new Supplementary Section 13.**

3. Based on the linear fluence dependence of the 3.7 THz mode observed in the low probe photon energy region, I expect that the phonon amplitude increases linearly with the fluence even up to the highest fluence in their measurement. If it were impulsive excitation, the oscillation will show a sin-like phase at least in the low fluence limit. On the other hand, if the potential energy surface of the lattice deformation Q of the phonon is directly coupled to the hidden order as shown in Fig. 1b, then the oscillation amplitude should deviate from the linear response as depicted in Fig. 1d. That is, the linear increase of the phonon amplitude in Fig. 4d and the model in Fig. 1 cannot compromise with each other. Instead, if the 3.7 THz mode were not directly coupled to the quadruple (or any other) order but the optical modulations due to the phonon oscillations should depend on the order parameter, then their observations could be consistent with each other. However, the origin/driving force of the new order still remains unclear.

This question can be well addressed based on the answer to the previous question. Fig. 1 represents a toy model with only one type of distortion. However, as we explained in the answer to the previous question, a more realistic model needs to include both Q_θ and Q_ϵ . Our simulation based

on the microscopic Hamiltonian including both coordinates shows that the oscillation amplitudes along both coordinates scale linearly with fluence up to F_c and then deviate from linearity, which is expected to be a general feature of any coherent phonon-induced dynamical transition (Fig. 1). A specific prediction of our model is that the deviation from linearity should be considerably more drastic for Q_ϵ than for Q_θ (Fig. 4b). As we explained in the paper, the fluence dependence of the 3.7 THz mode measured with different probe energies might seem inconsistent at first glance, but it directly reflects the prediction of the microscopic model. This is because a probe photon energy tuned to the α -peak is particularly sensitive to Q_θ , while a probe photon energy tuned to the β -peak is particularly sensitive to Q_ϵ . Since the eigenvector of the 3.7 THz QOCP has a finite projection along both coordinates, our model predicts that the deviation from linear fluence scaling above F_c observed for the 3.7 THz mode should be much more pronounced when the probe is tuned to the β -peak than when it is tuned to the α -peak. This prediction aligns well with our experimental observations (Fig. 4d).

However, we realize that the confusion may have arisen from the seemingly linear fluence dependence of the 3.7 THz mode with probe energies resonant with the α -peak. The explanation is that the nonlinear effects of Q_θ are relatively small and cannot be easily resolved experimentally. **We have now clarified this point in the caption of Fig. 4 to avoid any confusion.**

4. If the phonon mode is strongly coupled to the order parameter as discussed in Fig. 1, then the oscillation phase is also expected to deviate from the sine-like phase near F_c . Indeed, the reference #26 reported that the oscillation phase changes by 180 degrees depending on temperature. The temperature is not mentioned but figure S2 implies that some oscillation phase may change depending on fluence and/or photon energy. To understand better, it is desired to discuss the fluence dependent evolution of the oscillation phases.

The Reviewer raised questions regarding the temperature, probe energy, and fluence dependence of the QOCP phase. We will address each of these issues in turn.

First, we discuss the temperature dependence of the QOCP phase. As reported in Ref. 26, the π -phase flip stems from a static antipolar distortion of the apical oxygen atoms developing around T_{QO} . Therefore, this phase flip should also manifest in our experiment. As expected, our temperature-dependent 0.3 eV pump experiment shows a similar π -phase flip around T_{QO} with probe energies resonant with either α - or β -peaks (Fig. R4a), in agreement with Ref. 26.

We next comment on the probe energy dependence of the QOCP phase. For an electronic resonance whose energy is strongly modulated by phonons, the phonon oscillation reaches its peak amplitude where the first-derivative (slope) of the electronic resonance is largest [*Nat. Commun.* **13**, 98 (2022), *Nature* **609**, 282–286 (2022)]. Moreover, the phonon oscillation should have opposite phases below and above the peak position, due to the phonon modulation of the position of the electronic resonance (Fig. R4b inset). By comparing the phonon time traces (Fig. R4b) and

Fig. R4. **a**, Temperature dependence of the 3.7 THz phonon phase measured with a pump fluence of 15 mJ/cm² and probe at 1.55 eV and 0.69 eV. The dashed line marks T_{QO} . **b**, Probe energy dependent background-subtracted reflectivity transient spectrum measured with a pump fluence of 12 mJ/cm² at 80 K. Dashed lines denote the positions of α - and β -peaks. Inset shows if phonon modulates the position of the electronic resonance, the measured phase below and above the peak will be opposite. **c**, Fluence dependence of the QOCP amplitude from the simulation. Gray shaded area marks the fluence range where the final metastable state is a different quadrupolar order, while white shaded area marks the fluence range where the final metastable state is the initial quadrupolar order. **d**, Fluence dependence of the 3.7 THz phonon phase measured at select probe energies at 80 K.

the optical conductivity spectra, it is apparent that not only is the phonon amplitude large when the probe energy coincides with the rising edge of the two peaks and vanishes when there is a peak or dip in the optical spectrum (i.e. at 1.1, 1.6, and 2.1 eV), but a π -phase difference between the measured phonon phase below and above the α -peak position can also be resolved. This phenomenon, signifying the position of electronic resonances, has been observed elsewhere [*Nature* **609**, 282–286 (2022)].

Moving on to the fluence dependence of the QOCP, we need to first emphasize that the phonon phase is ill-defined for an anharmonic potential, especially in the vicinity of a photo-induced phase transition as in our case. There are two factors preventing us from drawing a definitive conclusion from the experimental results. First, as pointed out by the Reviewer, the phonon phase should deviate from $\pi/2$ when the QO switch occurs. However, due to the dynamical nature of the transition, the phase shift only occurs after the switch is completed, taking at least half of the

phonon period. On the other hand, the initial direction of the phonon oscillation, unlike the cases of changing temperature or probe energy, remains unchanged as the initial static condition is independent of pump fluence. In other words, the phonon phase evolves as a function of time, making a strict definition of phonon phase challenging. Second, as discussed in Supplementary Section 9, the system may transiently traverse multiple PES minima when the pump fluence is high. Moreover, as shown by a more comprehensive dynamical simulation using our microscopic model, the final metastable state, which is either the initial or a different QO, sensitively depends on the pump fluence (Fig. R4c white and gray regions). Therefore, in the gray fluence range where the final state is a different QO, the phonon phase will deviate from the low fluence regime where no transition occurs. However, in the white regions where the transition to a different order is only transient and the system finally relaxes back into the original QO, the long-time phonon phase should still be nearly identical to that of the low fluence regime. This complexity precludes obtaining a meaningful phonon phase. Due to these difficulties and our limited fluence sampling, we refrain from drawing any quantitative conclusions from our experimental data (Fig. R4d). **We have now reorganized and consolidated these discussions in a new Supplementary Section 15 to enhance clarity and organization.**

5. Figure 3b shows temperature dependence of the phonon amplitude. Their data at room temperature somehow have a very large error bar, resulting in practically no change above $T_{QO} \sim 260$ K. However, the reference #26 shows that the oscillation amplitude increases above 260 K. It is desired to explain the difference between these two data. It seems that the error bar becomes relatively larger at higher fluence according to Fig. 3. So, temperature dependence measured at lower pump fluence should be helpful to understand the evolution better. In addition, temperature dependent oscillation amplitude at different probe energies (near peak alpha and beta) is also desired to clarify the least coupling of the phonon mode with the alpha peak.

The Reviewer has raised four questions and we will address each of them in detail. We would like to first acknowledge the difference in experimental conditions between our work and Ref. 26. The experiment in Ref. 26 used a pump with a photon energy (1.55 eV) above the bandgap and a fluence range within the linear response regime ($F < 1$ mJ/cm²). In contrast, our experiment employed a pump with a photon energy (0.3 eV) below the gap, enabling the highest fluence up to $F > 25$ mJ/cm². To this end, Ref. 26 mainly explores the ground state properties with a relatively weak perturbation, while our work investigates the possible switch of QO in the strongly-driven out-of-equilibrium regime. Given the drastic difference in pump photon energy and fluence range, it is expected that different results would be observed in the two studies.

Next we comment on the temperature dependence of the 3.7 THz mode at a lower pump fluence. We need to first emphasize that a pump fluence smaller than F_c cannot realize the ultrafast switch to a hidden QO and the measured phonon amplitude cannot represent the maximal static JT

distortion where the potential energy is minimized. Our temperature dependence measurements in the main text were conducted with a pump fluence slightly larger than F_c at 80 K, ensuring that the dynamical transition occurs at all measured temperatures. To investigate the temperature dependence of the 3.7 THz mode at lower pump fluences, **we performed new experiments with a probe at 1.55 eV and a weaker pump fluence of 10 mJ/cm², which is larger than F_c at $T > 150$ K and close to (albeit smaller than) F_c at $T < 150$ K (Fig. R5a).** We find that the phonon amplitude shows a similar temperature dependence as that measured with a pump fluence of 15 mJ/cm² (Fig. R5c), confirming that as long as the pump fluence is close to F_c , our results shown in the main text should qualitatively hold. However, performing experiments with pump fluences similar to those in Ref. 26 ($F < 1$ mJ/cm²) would not only provide very limited information about the ultrafast change in QO but also be technically challenging given the limited signal-to-noise ratio of the 1 kHz laser used in our work compared to the 250 kHz laser used in Ref. 26. **These new results and discussions have been added to Supplementary Section 6.**

Fig. R5. **a**, Temperature dependent FFT spectra for 1.55 eV probe and 0.3 eV pump at 10 mJ/cm². Curves are offset vertically for clarity. **b**, Temperature dependent FFT spectra for 0.69 eV probe and 0.3 eV pump at 10 mJ/cm². Curves are offset vertically for clarity. **c**, Temperature dependence of the 3.7 THz phonon amplitude corrected for the pump-induced temperature rise obtained at different conditions as labelled in the legend. The thick colored lines are guides to the eye.

The third question pertains to the temperature dependence of the QOCP with a probe energy resonant with the α -peak, in addition to our current temperature dependence measured with a probe energy resonant with the β -peak. To address this, **we conducted new temperature dependent measurements with 0.3 eV pump at 10 mJ/cm² and 0.69 eV probe, which is resonant with the rising edge of the α -peak (Fig. R5b).** The phonon amplitude in general shows a similar temperature dependence as that measured with a probe energy of 1.55 eV, albeit with a lower

signal-to-noise ratio (Fig. R5c). This is expected in our microscopic model, because probe energy resonant with the α - and β -peaks is sensitive to Q_{θ} and Q_{ϵ} , respectively. Since both of them exhibit an upturn at T_{Q0} , their temperature dependence should be qualitatively similar. **These new results and discussions have been added to Supplementary Section 6.** We also note that any measurement with probe energies exactly resonant with α - and β -peaks is extremely challenging, because as addressed in the response to the previous question, the phonon amplitude is very small at these energies (see the reflectivity transients measured at a probe energy of 1.04 eV and 2.07 eV in Fig. 1).

Fig. R6. **a**, Temperature dependent FFT spectra for 1.55 eV probe and 1 eV pump at 3 mJ/cm². Curves are offset vertically for clarity. **b**, Temperature dependent FFT spectra for 0.69 eV probe and 1 eV pump at 3 mJ/cm². Curves are offset vertically for clarity. **c**, Temperature dependence of the 3.7 THz phonon amplitude corrected for the pump-induced temperature rise obtained at different conditions as labelled in the legend. The thick colored lines are guides to the eye.

Finally, we comment on whether the phonon amplitude increases above T_{Q0} . As previously shown, the data obtained with a 0.3 eV pump at various fluences and probe energies do not suggest a clear upturn above T_{Q0} , but an upturn of phonon amplitude above T_{Q0} with temperature was observed in Ref. 26. This may indicate that the pump photon energy, rather than the pump fluence or probe energy, is the primary cause of this distinction. To test this speculation, **we performed new temperature dependent measurements with an above-gap pump at a photon energy of 1 eV and $F = 3$ mJ/cm², which is nearly resonant with the α -peak, and probed with energies resonant with both α - and β -peaks.** Interestingly, we observe a more evident upturn of phonon amplitude above T_{Q0} with 1 eV pump compared to the 0.3 eV pump at both probe energies, resembling the results reported in Ref. 26 (Fig. R6). This may indicate that the pump photon energy plays a more important role in generating this behavior. This may be understood because subgap

pumping and above-gap pumping have different excitation mechanisms. The phonon amplitude with a subgap pump (ISRS) is determined by $d\text{Re}(\epsilon)/dE$, where ϵ is the complex dielectric constant and E is the photon energy, whereas the phonon amplitude with an above-gap pump (DECP) is determined by $\text{Im}(\epsilon)/\omega_{\text{phonon}}$, which is related to the electronic absorption [*PRB* **65**, 144304 (2002), *PRB* **94**, 184307 (2016)]. These two tensors may have different temperature dependence. A clear answer to this question needs more comprehensive pump energy and fluence dependent measurements, which are beyond the scope of the current work. However, due to these diverse observations, **we have explicitly mentioned the upturn of the phonon amplitude above T_{Q0} in the caption of Fig. 3 and directly referred to Ref. 26. Additionally, we included the results of the 1 eV pump in the same Supplementary Section 6.** It is also important to note that the solid lines shown in Figs. 3b and 3c are not fits but simulation results from the minimal microscopic model. The temperature dependence of the non-coupled part of the 3.7 THz phonon, i.e. the phonon amplitude above T_{Q0} , is not captured by the microscopic model. **We have added this clarification in the Fig. 3 caption.**

6. In the simulation, they initiated Q_{θ} and/or Q_{ϵ} instead of the phonon mode obtained from their DFT calculations and obtained the oscillation amplitude. What are the oscillation frequencies observed in the simulation? Do they result in the same oscillation frequency? I suppose that Q_{θ} and Q_{ϵ} displacement cannot oscillate at the observed phonon frequency. Simulations for the observed phonon mode depending on fluence are desired instead of the Q_{θ} and Q_{ϵ} .

We would like to first emphasize that our study incorporates both microscopic model simulations as well as DFT simulations. Similar microscopic models have been used in a large variety of theoretical works studying strongly correlated materials with d^1 [*Phys. Rev. B* **82**, 174440 (2010)], d^2 [*Phys. Rev. B* **84**, 094420 (2011)], d^4 , and d^5 fillings [*Phys. Rev. Lett.* **122**, 057203 (2019), *Phys. Rev. X* **10**, 031043 (2020)]. More specifically, this single-ionic model has been applied to understand the L-*Pbca*-to-S-*Pbca* structural phase transition in Ca_2RuO_4 , which involves the static Q_{θ} distortion [*Phys. Rev. Lett.* **121**, 067601 (2018)]. Therefore, this model is particularly suitable to understand Ca_2RuO_4 . Although intuitive, it is not first-principles and cannot accurately predict the mode frequencies. On the other hand, the DFT calculations can qualitatively predict phonon frequencies and eigenvectors but are limited in providing a microscopic understanding of the interplay between different degrees of freedom. Recognizing these drawbacks and the complementary nature of both methods, we combine them to gain a quantitative understanding of the physics in Ca_2RuO_4 . **We have also added a new Supplementary Section 13 to elaborate on the relationship between the 3.7 THz phonon and Q_{θ} and Q_{ϵ} , bridging the microscopic model and DFT simulation results.**

As addressed in the responses to questions 2 and 3, Q_{θ} and Q_{ϵ} are not actual phonon eigenmodes of Ca_2RuO_4 but two eigen-deformations of an octahedron. For the QOCP, which can map onto Q_{θ} and Q_{ϵ} , they both have the same frequency at 3.7 THz. Similarly, for the 5.7 THz QO-uncoupled mode, which can also map onto Q_{θ} and Q_{ϵ} but with smaller components, these two modes will also have the same frequency at 5.7 THz. Therefore, they will always have the same frequency as the phonon mode they derive from. **We have added these discussions in Supplementary Section 13.**

We agree with the Reviewer that a first-principles simulation for the fluence dependence of the 3.7 THz phonon or a time-dependent simulation to predict the QO switch will be valuable. However, this cannot be achieved solely through a static DFT calculation. Achieving these would require state-of-the-art methods such as time-dependent DFT (TDDFT) and time-dependent dynamic mean field theory (TD-DMFT) calculations. These novel approaches are still in their infancy and pose significant challenges, not only for application to strongly correlated systems like Ca_2RuO_4 but also for analyzing the long timescales required to access phonon dynamics. While we acknowledge the potential of such calculations, we believe they are beyond the scope of the current work. Nonetheless, **we have added a comment in the outlook section of the main text, suggesting further theoretical investigations using advanced time-dependent theories.**

I believe that their observations could indeed be related with some transient ordering. With above mentioned points, however, I think it is not clarified well enough why the ordering has to be the quadruple order. Therefore, I cannot recommend the publication until they clarify all the points.

We thank the Reviewer for agreeing that our observations are related to transient ordering. We would like to further summarize why it has to be transient quadrupolar ordering in the following logic chain as indicated in the main text:

- The amplitude of the 3.7 THz phonon exhibits an amplitude saturation at $F_c = 15 \text{ mJ/cm}^2$ at 80 K where the amplitudes of the other modes scale quasi-linearly with pump fluence. This dichotomy demonstrates that the anomaly cannot arise from absorption saturation and potentially suggests a dynamical phase transition;
- There are only three thermal phase transitions in Ca_2RuO_4 , namely the metal-to-insulator transition (MIT), quadrupolar ordering transition, and antiferromagnetic transition. The frequency of the 3.7 THz mode barely changes with fluence, in contrast to its strong temperature dependence, demonstrating the transition is nonthermal; The tiny change in the optical conductivity spectrum falsifies the possibility of a nonthermal MIT, leaving a nonthermal QO or magnetic order transition as the only possibilities.

- The temperature dependence of the 3.7 THz phonon amplitude and F_c show an abrupt change at T_{QO} but no change at T_{N} , supporting the picture of dynamical switching to a hidden QO state rather than to a hidden antiferromagnetic state.
- The fluence dependence of the QOCP amplitude can be qualitatively understood with our microscopic model. It also predicts a peculiar probe energy dependence which is consistent with our experiments.

We note that although more exotic transient ordering might be possible, our model based on the phenomenology provides the most simple explanation since QO is already reported to exist in equilibrium.

As a minor point, it is desired to mention important experimental conditions in more details, such as temperature and photon energy, in Fig. 4, Fig. S2.

We thank the Reviewer for this reminder. **We have now clarified the pump fluence, probe photon energy, and temperature in the captions of Figs 2, 3, 4, S2, S4, and S6.**

Response to Reviewer #2

The authors have presented a very comprehensive and solid study of the coherent phonon induced ordered state in Ca₂RuO₄. I find the results very interesting and important, since the observation of the hidden order in Ca₂RuO₄, as the authors written in the text, was quite challenging to be realized. Thus, I recommend to accept this paper for publication in Nature communications after the following suggestions are considered:

We thank the Reviewer for recommending our paper for publication in Nature Communications. We also thank the Reviewer for providing valuable comments. Below please find our response to each comment. **The resulting actions taken are summarized and noted in bolded text.**

1. The hidden quadrupolar order observed by the authors was attributed to an orbital ordered state occurs at the tetragonal phase transition point T_{QO}=260K. If I understand correctly, below T_N=110 K where the long range magnetic order forms, the system should correspond to a dipolar order which has orthorhombic symmetry. Then why do the observed features at T=80 K (below T_N) almost identical to those at T=220K (between T_N and T_{QO})?

Strictly speaking, the quadrupolar order (QO) is not an orbital (*L*) order but a pseudospin (*J*) order in the presence of spin-orbit coupling. For *d*⁴ (*J*=0) systems specifically, a quadrupolar ordering that breaks rotational symmetry (cubic-to-tetragonal phase transition) should theoretically precede the dipolar magnetic phase, which eventually breaks time-reversal symmetry [*Phys. Rev. Lett.* **122**, 057203 (2019)]. However, the ensuing magnetic transition occurring at a lower temperature *does not* necessarily accompany a tetragonal-to-orthorhombic phase transition. Since no more rotational symmetry breaking occurs, the coupling between the QO and lattice should exhibit no anomaly below and above T_N and we therefore should not expect changes in the phonon behavior.

The specific case for Ca₂RuO₄ is a bit more subtle. The octahedron exhibits tetragonality instead of cubicity even above T_{QO}, so only a subtle lattice change accompanies the quadrupolar ordering without explicit symmetry breaking. Although difficult to detect with X-ray diffraction [*Phys. Rev. B* **63**, 174432 (2001)], this subtle change manifests as a QO-coupled phonon phase flip as captured by a recent coherent phonon spectroscopy study [*Phys. Rev. B* **98**, 161115 (2018)]. Our experiments further employ this coupled phonon to drive and detect a light-induced QO transition. **We have now included all these clarifications in Supplementary Section 1.**

2. Section 1 in the supplementary materials is a little bit confusing and needs to be improved. For instance:

We apologize for causing any confusion. **We have now rewritten and reorganized the entire Supplementary Section 1 to better illustrate the microscopic picture of the QO in Ca₂RuO₄.**

a) what do the two shapes on t_{2g} orbitals in Fig. S1 represent?

The four balls represent the four electrons in the sixfold degenerate t_{2g} manifold. The electron number is important since it will determine the pseudospin energy spectrum in the presence of spin-orbit coupling. **We have now added more comprehensive explanations in the caption of Fig. S1 to elaborate the energy level evolution.**

b) Notations of “mJ” with capital “J” in the figure while it is small “j” in the text.

We apologize for the inconsistency. **We have changed all the notations in Fig. S1 so that the context and the figure legends are consistent.**

c) “Due to quasi-2D XY-type fluctuations, the magnetic ordering temperature T_N is lowered [11]”, I don’t think “T_N is lowered” “Due to quasi-2D XY-type fluctuations” is concluded by Ref.11. What Ref.11 delivers was the magnetic order should coincide with the lattice symmetry: quadrupolar order for tetragonal lattice and dipolar order for orthorhombic lattice.

We would like to clarify that we directly quoted the text from Ref. [11] without any interpretation. The relevant passage from Ref. [11] reads verbatim: “*While the cubic symmetry may be broken at finite temperature T_{JT}, long-range magnetic order is delayed due to XY-type phase fluctuations; therefore, T_m and T_{JT} are separated in quasi-2D J_{eff}=0 systems.*”

We think what the Reviewer refers to is the first section of the same reference, where in a J_{eff}=1/2 system the development of magnetic order accompanies an orthorhombic distortion. However, the physics discussed here is more directly related to the last section titled *Spin-nematic order in J_{eff}=0 systems*. **To avoid any possible misinterpretation, we have rephrased this sentence more directly as “However, in the quasi-2D J=0 system, owing to XY-type fluctuations, the magnetic ordering is suppressed to a temperature T_N — below the quadrupolar ordering temperature T_{QO}” and clearly cited the last section of the reference.** -

Response to Reviewer #3

The authors describe an experiment in which below-gap, mid-infrared light is used to impulsively excite various Raman-active modes in the Mott insulator $\text{Ca}_2\text{Ru}_2\text{O}_4$. The anomalous dependence of one phonon amplitude on pump fluence, temperature, and probe photon energy suggests that the material has transitioned dynamically to a thermally-inaccessible quadrupolar order. The stark disparity in the behavior of the quadrupolar order coupled phonon (QOCP) compared to the other observed coherent modes supports the authors' interpretation. While the paper does not introduce a novel approach to detecting quadrupolar orders (QOs), it does demonstrate how a transition among different QOs can be induced and experimentally measured.

In principle, I think that the idea is novel and that the model is solid. However, I have both major and minor concerns regarding the paper that the authors should address before I can recommend it for publication.

We thank the Reviewer for finding our work novel and solid. We also thank the Reviewer for providing valuable comments. Below please find our response to each comment. **The resulting actions taken are summarized and noted in bolded text.**

1. One of the most critical observations is that the QOCP amplitude saturates with fluence, while the amplitude of the other modes scales linearly with it. Regrettably, the figure (Fig. 2e) intended to confirm this behavior is confusing to me. Specifically, I believe that overlaying the data for all other phonons (Fig. 2e, right) could be misleading. Although I think a linear fit is appropriate for the 5.7 THz phonon, the fluence dependence of the 6.1 and 7.5 THz phonons appears to exhibit a similar, although less evident, saturation behavior at approximately 17.5 mJ/cm^2 .

To unequivocally demonstrate that the anomalies concern only the 3.7 THz phonon, I propose the following suggestions:

1.1 - Fig. 2e should display the fluence dependence for all phonons separately, similar to what is done for the 3.7 THz phonon. If possible, more data points (finer fluence steps) should be measured to differentiate between noise on top of a linear dependence and an actual saturation behavior for the other phonons.

We apologize for any confusion and appreciate the constructive suggestions. As suggested by the Reviewer, we have made changes to Fig. 2e (Fig. R7) to only display the fluence dependence for the 3.7 THz, 5.7 THz, and 7.5 THz phonons measured with a probe energy at 1.55 eV. This helps avoid potential misleading comparisons between phonons measured at different probe energies.

Fig. R7. Pump fluence dependence of the amplitude of the phonons probed at 1.55 eV. All the data are normalized by the 3.7 THz phonon amplitude measured at $F = 15 \text{ mJ/cm}^2$. Thick lines are guides to the eye.

To comprehensively understand whether the phonon amplitude anomaly is unique to the 3.7 THz mode, we investigate the pump fluence dependence of every mode measured over a wide range of probe energies resonant with either the d_{xy} - $d_{xz/yz}$ transition or the $d_{xz/yz}$ - $d_{xz/yz}$ transition (Fig. R8). All modes are now plotted separately as suggested by the Reviewer, allowing for thorough examination and cross-checking of the fluence dependence of different modes at various probe energies. The 5.7 THz mode observed at all the probe energies consistently shows a quasi-linear dependence on the pump fluence, highlighting its QO-uncoupled nature. Although the 7.5 THz mode only appears at $d_{xz/yz}$ - $d_{xz/yz}$ -resonant probe energies, it also shows clear linear dependence on pump fluence, in contrast to the 3.7 THz mode in the same probe energy range. These observations demonstrate that both the 5.7 THz and 7.5 THz modes are not quadrupolar-order-coupled phonons (QOCs).

The 6.1 THz, 9.1 THz, and 9.7 THz modes can only be resolved at probe energies resonant with the d_{xy} - $d_{xz/yz}$ transition. They all exhibit quasi-linear fluence dependence within the fitting error bars. There might be a small deviation from linearity at $F > 22 \text{ mJ/cm}^2$, but the raw FFT spectra (Fig. S16) still indicate an increase with fluence without saturation. Therefore, the deviation may result from the imperfect Lorentzian fits. However, since the 3.7 THz QOC also does not exhibit clear deviation from linearity at these probe energies, it remains possible that these modes are

QOCP and they might exhibit nonlinear pump fluence dependence if they could be observed with $d_{xz/yz}$ - $d_{xz/yz}$ -resonant probe energies.

In summary, as we claimed in the main text, *the amplitudes of all the other five modes continue to scale quasi-linearly up to $F = 25 \text{ mJ/cm}^2$ at all the probe energies where they can be observed.* This supports our main conclusion that the 3.7 THz mode is a QOCP, while the 5.7 THz and 7.5 THz modes are not. However, we cannot definitively determine whether the 6.1 THz, 9.1 THz, and 9.7 THz modes are QOCPs, necessitating further research. Regardless, our key finding that the saturation of a coupled phonon amplitude is indicative of a QO switch remains robust, even if more than one mode is coupled to the quadrupolar order (QO). **We have included these discussions and Fig. R8 in Supplementary Section 11 and revised our statement regarding whether these phonons are QOCP.**

Fig. R8. Pump fluence dependence of the amplitude of various phonons probed at select energies. All the data are normalized by the maximal 3.7 THz phonon amplitude measured at the corresponding probe energies. Thick lines are guides to the eye. The amplitude of the 9 THz and 9.8 THz modes are shown together because they cannot be clearly distinguished at most probe energies.

1.2 - The contents of Fig.3 should be presented in the supplementary materials for all the other phonons as well. Specifically, how does the amplitude of the other phonons change as a function of temperature?

We appreciate the Reviewer's suggestion, and in response, in Fig. R9, we present the fluence dependence of the 5.7 THz and 7.5 THz modes at 80, 130, 220, and 280 K as well as the temperature dependence of these modes measured at $F = 15 \text{ mJ/cm}^2$ with 1.55 eV probe, similar to the presentation for the 3.7 THz QOCP in Fig. 3. As expected, the quasi-linear pump fluence dependence persists at all the sampled temperatures for both modes, consistent with their QO-uncoupled nature. Moreover, the pump fluence dependence of the 5.7 THz mode measured at various temperatures almost follows a single line (Fig. R9a), supporting the temperature independence of the amplitude of the 5.7 THz mode (Fig. R9b). In contrast, the slope of the pump fluence dependence of the 7.5 THz mode, which reflects the photo-susceptibility at various temperatures, decreases with increasing temperature (Fig. R9c). This is also reflected in the temperature dependence of the 7.5 THz mode, where its amplitude decreases with increasing temperature, without any observable anomaly at T_{QO} (Fig. R9d), unlike the behavior observed for the 3.7 THz QOCP. These results provide additional confirmation that the 5.7 THz and 7.5 THz modes are not QOCP. **We have now included these discussions along with Fig. R9 in Supplemental Information Section 4 to provide a comprehensive analysis of the temperature dependence of the 5.7 THz and 7.5 THz modes.**

Fig. R9. **a**, Pump fluence dependence of the 5.7 THz phonon amplitude at select temperatures acquired with a 1.55 eV probe. All data are normalized to the maximal measured value. **b**, Temperature dependence of the 5.7 THz phonon amplitude measured at $F=15 \text{ mJ/cm}^2$ with a 1.55 eV probe. All data are normalized to the maximal measured value in **a**. The colored line is a guide to eye. **c** and **d** are the same as **a** and **b** but for the 7.5 THz mode.

1.3 – Also the contents of Fig.4d should be shown for all the other phonons. Does the fluence dependence remain linear at all other probe photon energies?

Most of the data I am requesting the authors to present should be readily available from the data sets they have already measured.

As addressed in response to comment 1.1, we have carefully examined the fluence dependence of the detected phonons at all the probe photon energies. Our findings reveal that the fluence dependence remains quasi-linear across all observed probe photon energies where the phonons can be detected. **We have now included the additional suggested data along with detailed discussions in Supplementary Section 11 to offer a comprehensive analysis of the fluence dependence of different phonons.**

2. The paper lacks clarity on how the driven Ag phonons affect and relate to the Eg tetragonal and orthorhombic deformations Q_{θ} and Q_{ϵ} are excited. The authors should explain whether the eigenvector of the 3.7 THz mode is mapped onto the Q_{θ} and Q_{ϵ} coordinates and how. Additionally, it is important to know whether the displacements of the other Ag modes at higher frequencies contribute to the Q_{θ} and Q_{ϵ} deformations and if this was considered in the calculations. In general, the authors should provide a better explanation of the relationship between the driven coherent phonons and the deformations used in their calculations.

We apologize for any confusion. As proposed by the Reviewer, the eigenvector of the 3.7 THz phonon mode can indeed be mapped onto the two octahedral eigen-deformations Q_{θ} and Q_{ϵ} . This mapping theoretically applies to all phonon eigenmodes. Considering a perfect octahedron with O_h point group symmetry, any inversion-symmetric distortion can be represented as a superposition of eigen-distortions respecting A_{1g} , A_{2g} , E_g , T_{1g} or T_{2g} symmetries. Therefore, these eigenmodes form a complete orthonormal basis for any type of distortion of an isolated octahedron. For the t_{2g} manifold electrons, only six different distortions respecting A_{1g} , E_g ($Q_{E\theta}$, $Q_{E\epsilon}$), and T_{2g} ($Q_{T\eta}$, $Q_{T\zeta}$, $Q_{T\zeta}$) symmetries are allowed to couple to the electrons, whose spatial configurations are displayed in Fig. S9 as reproduced here in Fig. R10 [*The Jahn-Teller Effect* (Cambridge University Press, Cambridge, 2006)].

In other words, any t_{2g} -electron-coupled phonon eigenmodes, regardless of their symmetries with respect to the entire lattice, is a superposition of the six octahedral eigenfunctions: $Q = a_A Q_A + a_{\theta} Q_{E\theta} + a_{\epsilon} Q_{E\epsilon} + a_{\zeta} Q_{T\zeta} + a_{\eta} Q_{T\eta} + a_{\zeta} Q_{T\zeta}$, where the a 's represent the ratio of different components. Since Q_A is fully symmetric and thus Jahn-Teller (JT) inactive, the leading-order bilinear JT coupling term can be expressed as: $g_E(a_{\theta} Q_{E\theta} \tau_{E\theta} + a_{\epsilon} Q_{E\epsilon} \tau_{E\epsilon}) + g_T(a_{\zeta} Q_{T\zeta} \tau_{T\zeta} + a_{\eta} Q_{T\eta} \tau_{T\eta} + a_{\zeta} Q_{T\zeta} \tau_{T\zeta})$, where

g 's are the JT coupling constants and τ 's are the electronic QO parameters. In transition metal oxides with only static E_g (tetragonal, orthorhombic) distortions like Ca_2RuO_4 , the T_{2g} terms can be omitted because g_T is significantly smaller than g_E [*Phys. Rev. B* **98**, 075138 (2018), *Phys. Rev. Lett.* **116**, 106401 (2016)]. Consequently, the coupling strength between the QO (τ) and any phonon mode is determined by $g(a_\theta Q_\theta \tau_\theta + a_\varepsilon Q_\varepsilon \tau_\varepsilon)$ (subscript “E” is dropped for simplicity). While these concepts have been introduced in Supplementary Section 7, **we now added a new paragraph in the main text to highlight the relationship between the driven coherent phonons and the octahedral deformations used in their calculations. Moreover, we reorganized a more detailed version of these important discussions in a new Supplementary Section 13.**

Fig. R10. Schematics of the eigenmodes of octahedron that can couple to t_{2g} electrons. **a**, Q_A , **b**, $Q_{E\theta}$, **c**, $Q_{E\varepsilon}$, **d**, $Q_{T\zeta}$, **e**, $Q_{T\eta}$, and **f**, $Q_{T\xi}$. The red balls represent O atoms at the apices of the octahedra and the gray balls represent Ru atoms

Having established the theoretical foundation, immediate questions are how the 3.7 THz QOCP is mapped to Q_θ and Q_ε and why the other modes are not coupled to the QO. To address these questions, **we utilized two methods to demonstrate how the 3.7 THz mode can be mapped onto Q_θ and Q_ε , i.e. obtain a_θ and a_ε . Furthermore, we applied the same analysis to the 5.7 THz phonon as a comparison, which is an unambiguously QO-uncoupled mode due to its consistent linear fluence dependence across the entire probe energy range.**

First, we qualitatively analyzed the components of Q_θ and Q_ε , i.e. a_θ and a_ε , by examining the amplitude and direction of the phonon eigenvectors. Ca_2RuO_4 is composed of in-plane corner-shared octahedra. As the octahedra undergo distortion, a collective tilting or rotation must follow to accommodate the bond length changes. By considering these geometric constraints, we imposed the required length change in the Ru-O bond of Q_θ and Q_ε and simulated the eigenvectors of Q_θ

Fig. R11. **a**, Schematic of the tetragonal distortion in the lattice. Only one layer of Ru-O octahedra is shown for clarity. Black arrows mark the desired displacement of the oxygen atoms in each individual octahedron. **b**, Schematic of the eigenvector of the tetragonal distortion in the lattice as marked by the yellow arrows, which are the sum of black arrows at each atomic position. **c**, Schematics showing the out-of-plane tilting of octahedra induced by the tetragonal distortion from the top and side views. Dashed lines indicate the tilting axes. **d-f**, Same as **a-c** but for the orthorhombic distortion.

and Q_{Eg} within the entire lattice after accommodating the bond length changes. Interestingly, we find that the eigenvector of the tetragonal distortion Q_{Eg} mainly exhibits an out-of-plane component, indicating a collective tilting of the octahedra along an in-plane axis (Figs. R11a-c). On the other hand, the eigenvector of the orthorhombic distortion Q_{Eg} mostly shows an in-plane component, suggesting a collective rotation of all the octahedra along an out-of-plane axis through the center Ru atoms (Figs. R11d-f). As a result, analyzing the direction of the eigenvectors allows us to roughly distinguish the components of Q_{Eg} and Q_{Eg} for different phonons. Another distinguishing feature between the two modes is the amplitude of displacement of the apical oxygens. The length of the apical Ru-O bonds remains unchanged in the Q_{Eg} mode, so the apical oxygen atoms only displace slightly following the rotation of the octahedron. However, the net displacement of the apical oxygen of the Q_{Eg} mode is significantly larger, due to the combination of tilting of the octahedron and the required length change for Q_{Eg} .

We now examine the eigenvectors of the 3.7 THz and 5.7 THz modes obtained from our DFT simulations (Supplementary Section 12) using the aforementioned two criteria. We find that the eigenvectors of the 3.7 THz mode displays a notably larger in-plane displacement of the planar oxygens and a larger displacement of the apical oxygens compared to the 5.7 THz mode. These two observations suggest a larger component of both Q_ε and Q_θ components in the 3.7 THz mode compared to the QO-uncoupled 5.7 THz mode. Given the larger projection on Q_ε and Q_θ , the 3.7 THz mode appears to be a more suitable candidate for QOCP than the 5.7 THz mode.

To more quantitatively obtain a_θ and a_ε , we calculated the length change in the four planar Ru-O bonds and two apical Ru-O bonds based on the DFT results. Theoretically, Q_θ induces a length change with the same amplitude (d_θ) and sign in the four planar Ru-O bonds, while Q_ε results in a length change with the same amplitude (d_ε) and opposite signs in the neighboring planar Ru-O bonds. Consequently, the net length change of two neighboring planar Ru-O bonds should be $d_\theta - d_\varepsilon$ and $d_\theta + d_\varepsilon$ when both distortions are present. The apical Ru-O bond change, on the other hand, predominantly originates from Q_θ . Since $d_\theta:d_\varepsilon = a_\theta:a_\varepsilon$, analyzing the length change of different Ru-O bonds provides a direct quantitative way to determine a_ε and a_θ . With identical Ca atom displacements for both modes, we find that the neighboring planar Ru-O bond length change ratio is roughly 1:2.36 and 1.29:1.32 for the 3.7 THz and 5.7 THz modes, respectively. This indicates that $a_\theta:a_\varepsilon = 1.68:0.68$ and $1.31:0.01$ for the two modes, respectively. In other words, the projection on Q_θ (a_θ) of the 3.7 THz mode is nearly $1.68/1.31 = 1.3$ times higher than that of the 5.7 THz mode. This is further corroborated by the length change in the apical Ru-O bond, with the 3.7 THz mode exhibiting nearly double the bond length change compared to the 5.7 THz mode. On the other hand, the projection on Q_ε (a_ε) of the 5.7 THz mode is nearly zero. The negligible composition of Q_ε in the 5.7 THz mode explains the absence of phonon amplitude saturation (which is most apparent in Q_ε) in the 5.7 THz mode.

Based on the aforementioned qualitative and quantitative analysis, we have demonstrated that both a_θ and a_ε are higher in the 3.7 THz mode. However, the formula $g(a_\theta Q_\theta \tau_\theta + a_\varepsilon Q_\varepsilon \tau_\varepsilon)$ indicates that the coupling strength to the QO is also determined by g in addition to a_θ and a_ε . Interestingly, our DFT simulation (Fig. S18c) has shown that the 3.7 THz mode induces a larger change in optical conductivity than the 5.5 THz mode at the probe energies sensitive to both Q_ε and Q_θ modes. This result indicates a larger g of the 3.7 THz mode, further supporting that the 3.7 THz mode is strongly coupled to the QO. As a result, our comprehensive analysis not only reveals how the phonons are projected onto Q_ε and Q_θ but also explains why the 3.7 THz mode is more strongly coupled to the QO. **We have now included all these discussions in the new Supplementary Section 13.**

3. The critical displacement (in pm) needed to transition to the hidden state should be estimated from both the experiments and calculations. This information can be valuable and could also be measured with the x-ray scattering experiments that the authors already considered.

As suggested by the Reviewer, we have estimated the change in the Ru-O bond length resulting from the phonon oscillation using the reflectivity transients. Our analysis reveals an atomic displacement of approximately 2 pm. This is in qualitative agreement with the difference in length between the planar and apical Ru-O bonds of approximately 3 pm - according to our theory, as the QO is switched from d_{xy} -dominated to $d_{xz/yz}$ -dominated, the bond length between the planar and the apical Ru-O bonds should also undergo a switch. The qualitative agreement between the estimated value and the real bond difference serves as evidence that the transient QO transition is indeed achievable at the critical pump fluence employed in our study. **We have added a new Supplemental Information Section 16 to elaborate on these findings.**

4. The paper states that the pump photon energy (0.3 eV) was chosen to be less than the gap (0.6 eV) and far away from phonon resonances. However, the latter information should be mentioned in the main text and not just in the supplementary material. It would also be helpful if the authors could provide a reference for or show the optical properties of the material at the pump wavelength, such as its absorption, reflectivity, transmission, etc.

We thank the Reviewer for raising this important point. Indeed, our pump photon energy is far from both the electronic and phononic resonances (Fig. R12). **As suggested by the Reviewer, we have included the latter point in the main text.**

We also agree with the Reviewer that figures showing optical properties of the material at the pump and probe wavelengths would be helpful. **To this end, we have included the optical conductivity and reflectivity spectra from 3 eV down to 0.1 eV as shown in Fig. R12, which covers both the pump and probe energies, in a new panel of Fig. S5.**

Fig. R12. Static optical conductivity and reflectivity spectra at 20 K. The 0.3 eV pump energy is marked by a vertical green line, while the probe energy range is shaded gray.

5. The same group has published and cited a previous work where they study the doublon-holon pair production in the same material and under the same excitation conditions [30]. Have the authors considered how such a simultaneous nonlinear process influences/affects the transition to the hidden state?

The Reviewer has raised an interesting point. To impulsively excite Raman active phonons with large amplitude while minimizing heating due to absorption, we choose the pump photon energy well below the Mott gap and far away from phonon resonances. However, our previous work [*Phys. Rev. Lett.* **128**, 187402 (2022)], which studies the electronic responses of the same material under identical conditions, reveals that despite the subgap pumping, doublon-holon pairs are still generated nonlinearly. Carrier excitation may transiently modulate the potential energy surface (PES), leading to a displacive excitation of phonons. However, the QOCP is predominantly excited by impulsive excitation based on two observations, which is justified in the new Supplementary Section 14. One observation is that the phase of the QOCP is close to $\pm\pi/2$, matching the expectation for an impulsively excited phonon and arguing against the displacive excitation scenario. The second observation is that the QOCP amplitude scales linearly with fluence before it saturates, which is inconsistent with a displacive excitation mechanism. This is because the amplitude of phonons excited displacively scales linearly with carrier density, but the carrier density exhibits a superlinear fluence dependence for subgap pumping either through multi-photon or quantum tunneling processes. Rather, our results are consistent with the impulsive Raman scattering process (Fig. R1). Therefore, the nonlinear carrier generation does not affect the predominantly impulsive excitation of QOCP and the QO switch. Moreover, we would like to note that even if the excitation were displacive, the QO switch can still be induced by the coupled phonon excitation and our main conclusion is still valid. This is because the QO switch is induced by the coherent phonon oscillation and is independent of the specific mechanism that drives the phonon at $t = 0$. In conclusion, nonlinear carrier generation does not affect our main conclusion that a coherent phonon can induce a switch to a hidden quadrupolar-ordered state. **We have addressed these concerns and included all the relevant discussions in a new Supplemental Section 17.**

6. In the supplementary material, Fig.S6e, Fig.S7a,b,c,d,e,f, Fig.S9c are missing.

We apologize for the missing figures. However, we have checked on our end and all the figures are displayed normally. In case the Reviewer continues to experience problems compiling the figures, we have attached these figures and their captions below for convenience.

Old Fig. S6; now Fig. S6e-g are incorporated into the new Fig. 3. Fig. R9 is included in Fig. S6.

Fig. S6. **a**, Temperature dependent FFT spectra for 1.55 eV probe and 0.3 eV pump at 15 mJ/cm². Curves are offset vertically for clarity. **b-d**, Fluence dependence of FFT spectra at three characteristic temperatures as denoted in panel **a**. The dashed lines are Lorentzian fits and the thick colored lines are guides to the eye. **e**, Schematics of the corresponding PES at select temperatures. **f**, Calculated temperature dependence of JT distortion Q_0 as marked in the bottom panel of **e**. **g**, Calculated temperature dependence of F_c . The dashed lines denote T_{Q_0} .

Old Fig. S7; New Fig. S9.

Fig. S7. Eigenmodes of the octahedron that can couple to t_{2g} electrons. **a**. Q_A , **b**. Q_{E0} , **c**. $Q_{E\epsilon}$, **d**. $Q_{T\xi}$, **e**. $Q_{T\eta}$, and **f**. $Q_{T\zeta}$. The red balls represent O atoms at the apices of the octahedra and the gray balls represent Ru atoms.

Old Fig. S9; New Fig. S11.

Fig. S9. **a**, JT distortion amplitude of E_g and T_{2g} distortions as a function of SOC in the d^1 electron configuration. **b**, E_g distortion amplitude as a function of SOC for d^1 and d^4 configurations. The orange bar denotes the value of λ for Ca_2RuO_4 . **c**, PES of d^1 and d^4 cases. All the parameters are the same except for the electron filling number N .

7. The authors should make the phonon amplitudes consistent for comparison. For instance, from Fig.3b, it is expected that the phonon amplitude at 15 mJ/cm^2 would increase by approximately 2.5 times going from $\sim 280\text{K}$ to $\sim 80\text{K}$. However, in Fig.3a, the phonon amplitude at these two temperatures and at 15 mJ/cm^2 is the same. It is unclear whether the data in Fig.3a is normalized. The authors should normalize the data when necessary and show the fit values otherwise to enable comparison between different data sets. The same applies to Fig.2e, where it is unclear whether the data on the right is normalized and the data on the left is not.

We apologize for the confusion and appreciate the Reviewer's suggestion. **We have now normalized all the phonon amplitudes to the value measured at $F = 15 \text{ mJ/cm}^2$ and $T = 80 \text{ K}$ in Fig. 3a (Fig. R13).** Indeed, we find the phonon amplitude at $F = 15 \text{ mJ/cm}^2$ and at 80 K is approximately 2.5 times larger than that at $F = 15 \text{ mJ/cm}^2$ and 280 K . **Similarly, we have also normalized all the data in Fig. 2e with the amplitude of the 3.7 THz phonon measured at $F = 15 \text{ mJ/cm}^2$ (Fig. R7) and all the data in Fig. 4d with the maximal phonon amplitude measured at each corresponding probe energy for better comparison.**

8. Fig.3a would benefit significantly from showing the data in the same plot without normalization. This would enable visualization of the changing critical fluence, maximum amplitude, and shape, as well as the stark increase in photo-susceptibility (the slope of the fluence dependence before saturation) for decreasing temperatures.

We thank the Reviewer for the suggestion. **We have now reorganized Fig. 3a as suggested. We have also divided Fig. 3b into two panels, so that the experimental temperature dependence of the QOCP amplitude and F_c overlaid with the simulation results can be individually and unambiguously displayed.** Here is a reproduction from the new manuscript for the Reviewer's reference.

Fig. R13. **Temperature dependence of the QOCP amplitude.** **a**, Pump fluence dependence of the 3.7 THz phonon amplitude at select temperatures acquired with a 1.55 eV probe. Solid colored lines are guides to eye. All the data are normalized to the value measured at $F = 15$ mJ/cm² and $T = 80$ K. **b,c**, Temperature dependence of the 3.7 THz phonon amplitude measured at $F = 15$ mJ/cm² with a 1.55 eV probe corrected for the pump-induced temperature rise (Supplementary Section 7) and temperature dependence of F_c . The theoretical temperature dependence of the QOCP amplitude and F_c calculated from our microscopic model are shown by the thick colored lines and are scaled vertically to match the experimental data (Supplementary Section 9). The dashed line denotes T_{QO} . The inset is a schematic showing that the phonon amplitude measured at $F \geq F_c$ is a proxy for the maximal static JT distortion where the PES minimum is located. Note that an upturn of phonon amplitude above T_{QO} is not captured by our minimal model but has been observed elsewhere [26].

Now from Fig. 3a, we can unambiguously visualize 1) the disappearance of saturation behavior at $T = 280$ K $> T_{QO}$; 2) the change in maximum phonon amplitude, a proxy for the maximal JT distortion where the potential energy is minimized when $T < T_{QO}$; 3) the change in F_c as a function of T ; and 4) the rapid increase in photo-susceptibility with decreasing temperature below T_{QO} .

REVIEWERS' COMMENTS

Reviewer #1 (Remarks to the Author):

The authors have successfully provided thorough explanations to all the raised points. I recommend it for publication.

Reviewer #2 (Remarks to the Author):

My comments and suggestions are addressed clearly, and incorporated into the manuscript appropriately.

In conclusion, I recommend the manuscript for publication in Nature Communications.

Reviewer #3 (Remarks to the Author):

I would like to recommend this paper for publication in Nature Communications. The authors have demonstrated a strong commitment to addressing all of my comments and concerns, and their revisions have significantly improved the quality and clarity of the manuscript. I believe that the paper is now well-prepared for publication and will make a valuable contribution to the field.

Reviewer #1 (Remarks to the Author):

The authors have successfully provided thorough explanations to all the raised points. I recommend it for publication.

We thank the Reviewer for providing valuable comments and recommending our paper for publication.

Reviewer #2 (Remarks to the Author):

My comments and suggestions are addressed clearly, and incorporated into the manuscript appropriately. In conclusion, I recommend the manuscript for publication in Nature Communications.

We thank the Reviewer for providing valuable comments and recommending our paper for publication.

Reviewer #3 (Remarks to the Author):

I would like to recommend this paper for publication in Nature Communications. The authors have demonstrated a strong commitment to addressing all of my comments and concerns, and their revisions have significantly improved the quality and clarity of the manuscript. I believe that the paper is now well-prepared for publication and will make a valuable contribution to the field.

We thank the Reviewer for providing valuable comments and recommending our paper for publication.